# A Versatile Diffusion Transformer with Mixture of Noise Levels for Audiovisual Generation

**Gwanghyun Kim**[1,†,*]  **Alonso Martinez**[2]  **Yu-Chuan Su**[2]

**Brendan Jou**[2]  **José Lezama**[2]  **Agrim Gupta**[3,*]  **Lijun Yu**[4,*]

**Lu Jiang**[4,*]  **Aren Jansen**[2]  **Jacob Walker**[2]  **Krishna Somandepalli**[2,†]

[1]Seoul National University  [2]Google DeepMind  [3]Stanford University  [4]Carnegie Mellon University

## Abstract

Training diffusion models for audiovisual sequences allows for a range of generation tasks by learning conditional distributions of various input-output combinations of the two modalities. Nevertheless, this strategy often requires training a separate model for each task which is expensive. Here, we propose a novel training approach to effectively learn arbitrary conditional distributions in the audiovisual space. Our key contribution lies in how we parameterize the diffusion timestep in the forward diffusion process. Instead of the standard fixed diffusion timestep, we propose applying variable diffusion timesteps across the temporal dimension and across modalities of the inputs. This formulation offers flexibility to introduce variable noise levels for various portions of the input, hence the term *mixture of noise levels*. We propose a transformer-based audiovisual latent diffusion model and show that it can be trained in a task-agnostic fashion using our approach to enable a variety of audiovisual generation tasks at inference time. Experiments demonstrate the versatility of our method in tackling cross-modal and multimodal interpolation tasks in the audiovisual space. Notably, our proposed approach surpasses baselines in generating temporally and perceptually consistent samples conditioned on the input. Project page: avdit2024.github.io

## 1 Introduction

Recent years have witnessed a remarkable surge in the development and exploration of multimodal diffusion models. Prominent examples include text-to-image (T2I) [37, 41, 54, 39], text-to-video (T2V) [17, 6, 12]. Despite notable advancements, generating sequences across multiple modalities, like video and audio, remains challenging and is an open research area.

Introducing a time axis to static data paves the way for diverse multimodal sequential tasks including cross-modal generation (e.g., audio-to-video), multimodal interpolation, and audiovisual continuation as shown in Fig 1. Each task can be further divided based on various input-output combinations of the modalities, leading to a number of conditional distributions. For example, with video data $x_0^{1:N}$ and audio data $y_0^{1:N}$ of length $N$, The complexity of configurations grows with tasks like audiovisual continuation, $p(x_0^{(n_c+1:N)}, y_0^{(n_c+1:N)}|x_0^{(1:n_c)}, y_0^{(1:n_c)})$, where $n_c$ is the input frame length used for conditioning, and multimodal interpolation, $p(x_0^{(n \in \mathcal{N}_x^c)}, y_0^{(n \in \mathcal{N}_y^c)}|x_0^{(n \in \mathcal{N}_x)}, y_0^{(n \in \mathcal{N}_y)})$, where $\mathcal{N}_x$ and

---

*Work done while at Google.

†Equal contribution. Corresponding author: Krishna Somandepalli ⟨ksoman@google.com⟩

38th Conference on Neural Information Processing Systems (NeurIPS 2024).

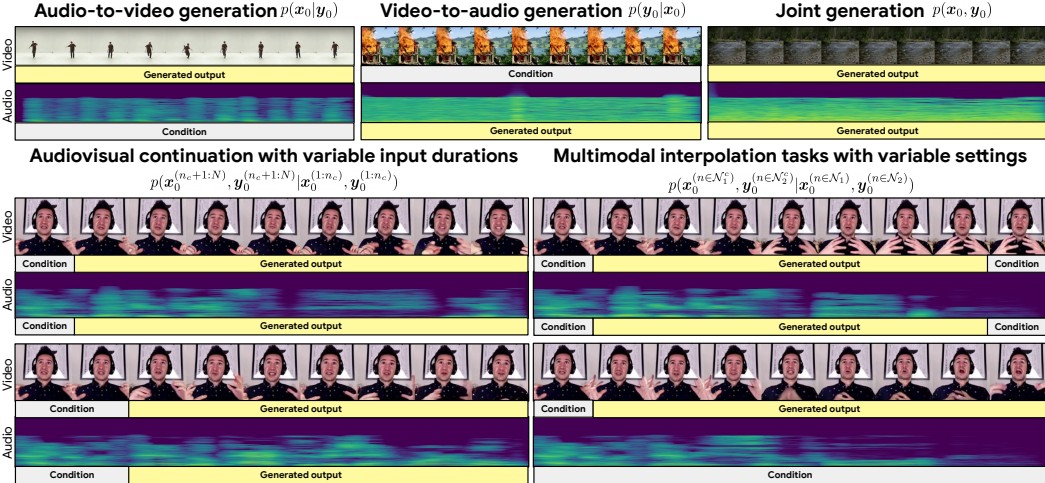

Figure 1: Our Audiovisual Diffusion Transformer trained with Mixture of Noise Levels tackles diverse AV generation tasks in a single model; see avdit2024.github.io for video demos.

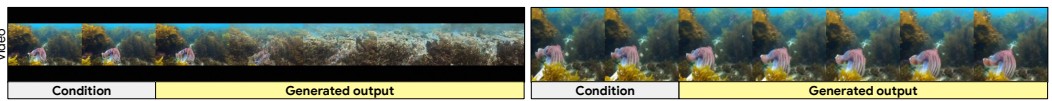

Figure 2: Comparing conditional inference for AV-continuation for MM-Diffusion (left) and Ours (right) on Landscape dataset. Our approach excels at generating temporally consistent sequences.

$\mathcal{N}_y$ are input index sets. Training separate models for each variation is expensive and impractical. A more efficient training approach would be to learn these conditional distributions in a single model without explicitly enumerating them, i.e., in a task-agnostic manner.

Unconditional diffusion models like MM-Diffusion [40] show potential for learning conditional distributions implicitly, but rely on inference adjustments [18, 40]. This limits performance, as seen in MM-Diffusion's struggle to generate temporally consistent sequences (see Fig. 2). While UniDiffuser [3] and Versatile Diffusion [51] offer methods for joint and conditional text-image distributions, effectively capturing temporal dynamics of audio and video remains an open challenge.

Here, we propose a multimodal diffusion framework that empowers a single model to learn diverse conditional distributions. This paves the way for a versatile framework for multimodal diffusion, tackling various generation tasks. Our core idea is that, applying variable noise levels across modalities and time segments [3] enables a single model to learn arbitrary conditional distributions. This formulation offers flexibility to train diffusion models with a *mixture of noise levels* i.e., **MoNL**, which introduces variable noise levels across various portions of the input. It has a number of advantages over previous approaches: it requires minimal modifications to the original denoising objective simplifying implementation, task-agnostic training, and support for conditional inference of a given task specification without any inference-time modifications.

We apply this approach for audiovisual generation by developing a diffusion transformer, **AVDiT**. To address the computational complexity of high-dimensional audio and video signals, we implement MoNL in the low-dimensional latent space learned by the MAGVIT-v2 [55] for video and the SoundStream [57] for audio. Importantly, the temporal structure in these latent representations enables us to apply variable noise levels. We also introduce a transformer-based network for joint noise prediction. Transformers are a natural choice for our implementation due to their proficiency to model multimodal data [25, 10] capturing complex temporal and cross-modal relationships.

We assess the capability of MoNL to model various distributions in the audiovisual space by evaluating cross-modal tasks (audio-to-video and video-to-audio generation), and conditioning on small portions (audiovisual continuation and interpolation tasks). For these tasks, we show that the AVDiT trained

---

[3] We use the term, "time-segment" to reference a single unit in time dimension of the inputs (e.g., frame in a video) during forward diffusion. Whereas, "timestep" or "diffusion timestep" refers to a single step in the process of adding noise during forward diffusion process.

with MoNL outperforms conventional methods including unconditional and conditional generation models, demonstrating the versatility of our task-agnostic framework as shown in Fig. 1. Notably, qualitative and quantitative evaluations highlight the ability of our framework to generate temporally consistent sequences, as illustrated in Fig. 2.

## 2 Background

**Diffusion Models for Multivariate series data:** Consider an example of a video diffusion model where the input is a sequence of image frames. In general, this task is modeling multivariate series data (i.e., image representations) of $d$-dimensions with $N$ elements (no. of image frames), henceforth referred to as *time-segments*. Thus, the multivariate series data, $\boldsymbol{x}_0 = \boldsymbol{x}_0^{1:N} \in \mathbb{R}^{N \times d} \sim q(\boldsymbol{x}_0)$ can be represented as a sequence of time-segments, where $x_0^n \in \mathbb{R}^d$ is the $n$-th time-segment and $d$-dimensional representation.

During the forward process of diffusion models [44, 16], the original data $\boldsymbol{x}_0$ is corrupted by gradually injecting noise in a sequence of $T$ timesteps. The noisy data $\boldsymbol{x}_t$ at time $t$ can be written as $\boldsymbol{x}_t = \boldsymbol{x}_t^{1:N} = \sqrt{\overline{\alpha}_t}\boldsymbol{x}_0 + \sqrt{1 - \overline{\alpha}_t}\boldsymbol{\epsilon}_x$. Here, $\boldsymbol{\epsilon}_x = \boldsymbol{\epsilon}_x^{1:N} \sim \mathcal{N}(\mathbf{0}, \mathbf{I})$ is Gaussian noise injected to the sequence and $\beta_t$ is the noise schedule, $\alpha_t = 1 - \beta_t$ controls the *noise level* at each step with $\overline{\alpha}_t = \prod_{i=1}^t \alpha_i$. Each noisy time-segment can be represented as $x_t^n = \sqrt{\overline{\alpha}_t}x_0^n + \sqrt{1 - \overline{\alpha}_t}\epsilon_x^n$. During the reverse process, the data is sampled through a chain of reversing the transition kernel $q(\boldsymbol{x}_{t-1}|\boldsymbol{x}_t)$ that is estimated by $p_{\boldsymbol{\theta}}(\boldsymbol{x}_{t-1}|\boldsymbol{x}_t) = \mathcal{N}(\boldsymbol{x}_{t-1}|\boldsymbol{\mu}(\boldsymbol{x}_t, t), \sigma_t^2\boldsymbol{I})$, where $\boldsymbol{\mu}(\boldsymbol{x}_t, t) = \sqrt{\alpha_t}(\boldsymbol{x}_t - \frac{1-\alpha_t}{\sqrt{1-\overline{\alpha}_t}}\boldsymbol{\epsilon}_{\boldsymbol{\theta}}(\boldsymbol{x}_t, t))$. The training objective is to learn a residual denoiser $\boldsymbol{\epsilon}_{\boldsymbol{\theta}}$ at each step as:

$$\min_{\boldsymbol{\theta}} \mathbb{E}_{t,\boldsymbol{x}_0,\boldsymbol{\epsilon}_x} \|\boldsymbol{\epsilon}_{\boldsymbol{\theta}}(\boldsymbol{x}_t, t) - \boldsymbol{\epsilon}_x\|_2^2, \tag{1}$$

where $t \sim \mathcal{U}(\{1, 2, \ldots, T\})$ is the diffusion timestep.

**Multimodal Diffusion Models:** Unconditional joint generation (generating all modalities simultaneously) and conditional generation (generating one modality conditioned on the rest) are commonly used for multimodal diffusion. Typically, separate models are trained for each task as described below:

**Diffusion models for joint generation.** For simplicity, let us assume two modalities $\boldsymbol{x}_0, \boldsymbol{y}_0$. The objective in joint generation is to model the joint data distribution, denoted as $q(\boldsymbol{x}_0, \boldsymbol{y}_0)$. To learn this, a joint noise prediction network, denoted as $\boldsymbol{\epsilon}_{\boldsymbol{\theta}}$ is defined by rewriting Eq. 1 as follows:

$$\min_{\boldsymbol{\theta}} \mathbb{E}_{t,\boldsymbol{x}_0,\boldsymbol{y}_0,\boldsymbol{\epsilon}_x,\boldsymbol{\epsilon}_y} \|\boldsymbol{\epsilon}_{\boldsymbol{\theta}}(\boldsymbol{x}_t, \boldsymbol{y}_t, t) - [\boldsymbol{\epsilon}_x, \boldsymbol{\epsilon}_y]\|_2^2, \tag{2}$$

where $(\boldsymbol{x}_0, \boldsymbol{y}_0)$ is a random data point, $[,]$ denotes concatenation, $\boldsymbol{\epsilon}_x, \boldsymbol{\epsilon}_y \sim \mathcal{N}(\mathbf{0}, \boldsymbol{I})$, and $t \sim \mathcal{U}(\{1, 2, \ldots, T\})$. Diffusion models trained with this objective can perform conditional sampling $q(\boldsymbol{x}_0|\boldsymbol{y}_0)$ using inference-time tricks [18, 40].

**Conditional training of diffusion models.** To learn conditional distributions, expressed as $q(\boldsymbol{x}_0|\boldsymbol{y}_0)$, a noise prediction network $\boldsymbol{\epsilon}_{\boldsymbol{\theta}}$ conditioned on $\boldsymbol{y}_0$ is adopted from Eq. 2:

$$\min_{\boldsymbol{\theta}} \mathbb{E}_{t,\boldsymbol{x}_0,\boldsymbol{y}_0,\boldsymbol{\epsilon}_x} \|\boldsymbol{\epsilon}_{\boldsymbol{\theta}}(\boldsymbol{x}_t, \boldsymbol{y}_0, t) - \boldsymbol{\epsilon}_x\|_2^2. \tag{3}$$

Separate conditional models need to be trained for every pair of modalities and input configurations.

## 3 Mixture of Noise Levels (MoNL)

We introduce a novel framework for learning a wide range of conditional distributions within multimodal data by using a mixture of noise levels. The key idea is to formulate the timestep $t$ (Eq. 1) that determines a noise level in the forward diffusion as a vector. Then, we present representative strategies for variable noise levels. We then show how conditional inference can be performed without additional training. Finally, assembling all these components, we present our versatile audiovisual diffusion transformer (AVDiT).

### 3.1 Variable Noise Levels across Modality and Time

Formally, let $M$ represent the number of modalities with sequence representations (latent spaces or raw data). Without loss of generality, assume the representations in each modality have $N$ time-

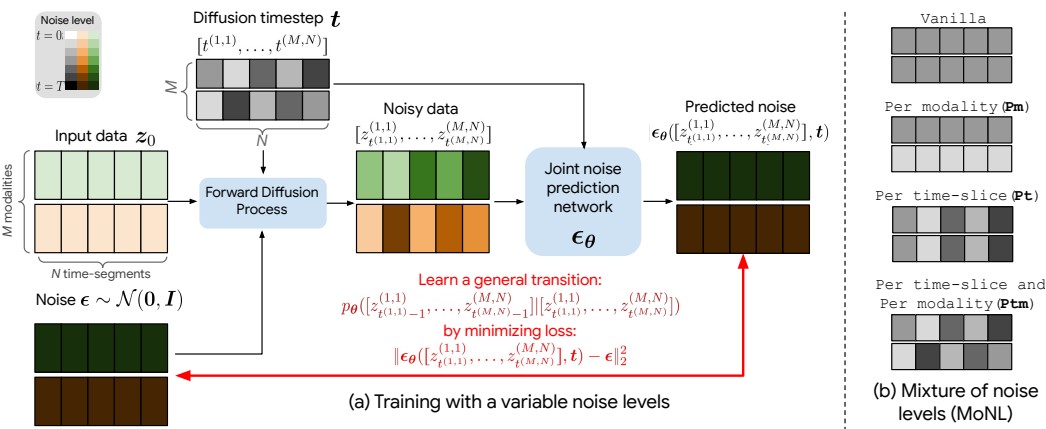

Figure 3: Overview of (a) diffusion training with variable noise levels per time-segment and per modalities, and (b) the mixture of noise levels. Intensity of the color is used to indicate variable noise levels applied to multimodal input. The original input data $z_0$ consists of $M$ modalities and $N$ time-segments. $z_0$ is then perturbed with noise $\epsilon$ per a noise level determined by a diffusion timestep vector $t$ to create noisy data $z_t$, which is input to the noise prediction network.

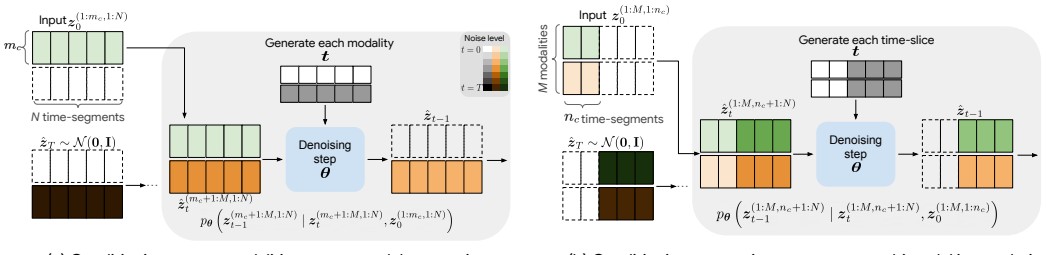

Figure 4: Illustration of the conditional inference in our framework for (a) cross-modal generation and (b) multimodal interpolation.

segments[4]. Let us further assume they have the same embedding dimension $d$ (which in practice can be achieved by projecting the noisy input from each modality to the desired dimension). The entire sequence can then be simplified as $z_0 \in \mathbb{R}^{M \times N \times d} = z_0^{(1:M, 1:N)} \sim q(z_0)$, where $z_0^{(m,n)} \in \mathbb{R}^d$ denotes the $n$-th time-segment of $m$-th modality. For reference, Sec. 2 represents two modalities of multivariate series data, $x_0^{1:N}$ and $y_0^{1:N}$, using this notation as $z_0^{(1,1:N)}$ and $z_0^{(2,1:N)}$, respectively.

We posit that training a single model to support learning arbitrary conditional distributions can be realized by using variable noise levels for each modality $m$ and time segment $n$ of the input space $z_0$. We introduce the diffusion timestep vector as $t = t^{(1:M,1:N)} \in \mathbb{R}^{M \times N}$ to match the dimensionality of the multimodal inputs, where each element $t^{(m,n)} \in [1, T]$ determines the timestep, and in turn the level of noise added to the corresponding element $z_0^{(m,n)}$ of the input $z_0$.

Recall (from Sec. 2) that in a unimodal case, the goal was to learn the transition kernel $q(x_{t-1}|x_t)$ parameterized by $p_\theta(x_{t-1}|x_t) = \mathcal{N}(x_{t-1}|\mu(x_t, t), \sigma_t^2 I)$. Analogously, by introducing a timestep vector $t \in \mathbb{R}^{M \times N}$, our goal is now to learn a general transition matrix between the various modalities and time-segments in $z_0$ at each step:

$$p_\theta([z_{t^{(1,1)}-1}^{(1,1)}, \ldots, z_{t^{(M,N)}-1}^{(M,N)}]|[z_{t^{(1,1)}}^{(1,1)}, \ldots, z_{t^{(M,N)}}^{(M,N)}]) \tag{4}$$

---

[4]In practice, this is rarely true; say, video and audio representations, the embedding dimension and temporal compression in the raw data or latent spaces can be vastly different. However, what we propose here can be generalized by keeping track of the frame-level correspondences between modalities

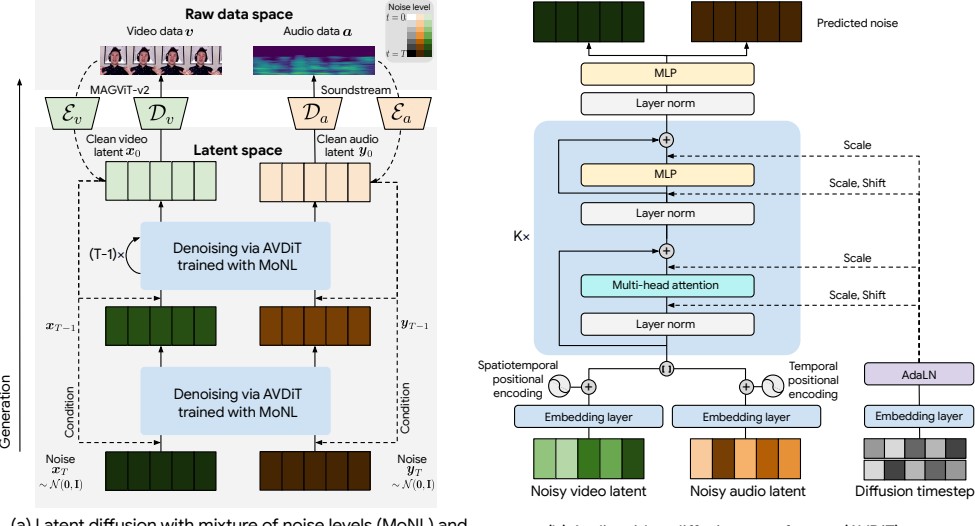

(a) Latent diffusion with mixture of noise levels (MoNL) and audiovisual diffusion transformer (AVDiT)

(b) Audio-video diffusion transformer (AVDiT)

Figure 5: Schematic of (a) the proposed approach, and (b) AV-transformer for joint noise prediction.

Then, for diffusion training, we draw a Gaussian noise sequence $\boldsymbol{\epsilon} = \boldsymbol{\epsilon}^{(1:M,1:N)}$. Each noise element $\epsilon^{(m,n)}$ is then added to the corresponding element of the original data $z_0^{(m,n)}$ with noise level determined by $t^{(m,n)}$ as follows:

$$z_{t^{(m,n)}}^{(m,n)} = \sqrt{\overline{\alpha}_{t^{(m,n)}}} z_0^{(m,n)} + \sqrt{1 - \overline{\alpha}_{t^{(m,n)}}} \epsilon^{(m,n)} \tag{5}$$

Then, the joint and conditional training objectives in Eqs. 2 and 3 can be generalized with a single noise prediction objective to learn the joint distribution $\boldsymbol{\epsilon_\theta}$ as follows:

$$\min_{\boldsymbol{\theta}} \mathbb{E}_{\boldsymbol{t}, \boldsymbol{z}_0, \boldsymbol{\epsilon}} \| \boldsymbol{\epsilon_\theta}([z_{t^{(1,1)}}^{(1,1)}, \dots, z_{t^{(M,N)}}^{(M,N)}], \boldsymbol{t}) - \boldsymbol{\epsilon} \|_2^2, \tag{6}$$

where $\boldsymbol{z}_0 \sim q(\boldsymbol{z}_0)$ is the multimodal input and $\boldsymbol{t}$ is the diffusion timestep vector.

## 3.2 Representative Stratgies for Variable Noise Levels

Using the generalized view of multimodal noise prediction described in Eq. 6, we now examine various strategies for variable noise levels during the forward diffusion. One can imagine an arbitrarily large number of timestep candidates in the vector space of $\boldsymbol{t}$ drawn as functions of time-segments of the multivariate series and modalities. Here, we explore four designs to create a mixture of noise levels as illustrated in Fig. 3(b). Let us assume we have final diffusion timestep vector for training, $\boldsymbol{t}_{ref} \in \mathbb{R}^{M \times N}$ where each element $t_{ref}^{(i,j)}$ is sampled from $\mathcal{U}(\{1, 2, \dots, T\})$,

- `Vanilla`: Same timestep is assigned to all the time-segments and modalities. This is analogous to performing joint learning as $t^{(m,n)} = t_{ref}^{(1,1)}$, and would be the straightforward way to extend the vanilla distillation approach for the multimodal case.
- `Per Modality (Pm)`: Variable timesteps are assigned for each modality, but all time-segments in a given modality have the same timestep as $t^{(m,n)} = t_{ref}^{(m,1)}$. This is expected to promote cross-modal generation tasks. This is a generalization of the UniDiffuser [3] approach for sequences.
- `Per Time-segment (Pt)`: Variable timesteps are assigned as $t^{(m,n)} = t_{ref}^{(1,n)}$ by keeping track of the corresponding time-segments across modalities. Intuitively, this should promote better temporal consistency.
- `Per Time-segment and Per-modality (Ptm)`: Variable timesteps are assigned for each time-segment and modality as $t^{(m,n)} = t_{ref}^{(m,n)}$. This would promote better temporal correspondence between modalities.

To enable learning a wide range of conditional distributions, we create a training paradigm where a timestep is uniformly randomly selected from the mixture. Specifically, we refer to this training paradigm as **MoNL**. A schematic of the overall training process is depicted in Fig. 3(a), with related pseudocodes in Algorithms 1 and 2 in the Appendix. Also, theoretical background on MoNL is detailed in Appendix G.

## 3.3 Conditional Inference

Once the general transition kernel $p_\theta$ is learned in Eq. 4, we investigate the model's ability to handle arbitrary conditional distributions. We achieve this by selectively injecting inputs during inference based on the task specification, i.e., clean (no noise) inputs for conditional portions with $t^{(m,n)} = 0$, and noisy inputs for generating desired portions of the input with the current diffusion step $t^{(m,n)} = t$.

Consider the case of cross-modal generation (Fig. 4(a)), to generate a sequence of $M - m_c$ modalities conditioned on $m_c \in (1, M)$ modalities, we set timestep elements of $M - m_c$ modalities as $t$ and those of $m_c$ conditioning modalities as 0, which achieves:

$$p_\theta\big(\boldsymbol{z}_{t-1}^{(m_c+1:M,1:N)}|\boldsymbol{z}_t^{(m_c+1:M,1:N)}, \boldsymbol{z}_0^{(1:m_c,1:N)}\big) \tag{7}$$

Similarly, for multimodal interpolation (Fig. 4(b)), to generate $N - n_c$ time-segments of all modalities jointly, conditioned on $n_c \in (1, N)$ time-segments, we set the timestep for the $N - n_c$ time-segments as $t$, and for the conditioning $n_c$ time-segments as 0, which achieves $p_\theta\big(\boldsymbol{z}_{t-1}^{(1:M,n_c+1:N)}|\boldsymbol{z}_t^{(1:M,n_c+1:N)}, \boldsymbol{z}_0^{(1:M,1:n_c)}\big)$. Unconditional joint generation is also possible by setting each timestep as the same $t$, to estimate the transition kernel, $p_\theta\big(\boldsymbol{z}_{t-1}^{(1:M,1:N)}|\boldsymbol{z}_t^{(1:M,1:N)}\big)$. Intuitively, our mixture of noise levels is analogous to self-supervised learning which bypasses the need for predefined tasks during training but enables a deeper understanding of multimodal temporal relationships. See also Sec. E for the discussion on classifier-free guidance in the Appendix.

## 4 Audiovisual Latent Diffusion Transformer (AVDiT)

Our model consists of two key components: (1) latent space representations from audio and video autoencoders, and (2) an Audiovisual diffusion transformer (AVDiT) for joint noise prediction.

**Latent Space Representations:** For a video of $1 + L_v$ frames, represented as $\boldsymbol{v} \in \mathbb{R}^{(1+L_v) \times H \times W \times C}$, we use MAGVIT-v2 [55], a causal autoencoder to achieve efficient spatial and temporal compression. MAGVIT-v2 results in a low-dimensional representation, $\boldsymbol{x}_0 \in \mathbb{R}^{(1+l_v) \times h \times w \times d_v}$, by a compression factor of $r_s = \frac{H}{h} = \frac{W}{w}$ in space and $r_{t_v} = \frac{L_v}{l_v}$ in time. Crucially, the use of causal 3D convolutions ensures that the embedding for a given frame is solely influenced by preceding frames, preventing flickering artifacts common in frame-level autoencoders.

For audio with $L_a$ frames, $\boldsymbol{a} \in \mathbb{R}^{L_a}$, we use SoundStream [57], a state-of-the-art neural audio autoencoder. We use the latents $\boldsymbol{y}_0 \in \mathbb{R}^{l_a \times d_a}$ prior to quantization as audio latents, a compression rate of $r_{t_a} = \frac{L_a}{l_a}$ in time. The *time-segments* in our formulation refer to the $1 + l_v$ and $l_a$ temporal dimensions in the video and audio latent spaces respectively.

**Audiovisual Transformer for Joint Noise Prediction:** Transformers [48] are a natural fit for multimodal generation as they can: (1) efficiently integrate multiple modalities and their interactions [58, 10], (2) capture intricate spatiotemporal dependencies [8, 5], and (3) have shown impressive video generation capabilities [12, 25]. Inspired by these benefits, we introduce AVDiT, a noise prediction network for latent diffusion as described in Fig. 5. AVDiT utilizes the timestep embedding similar to the condition signal used in W.A.L.T [12]. The Transformer first processes the timestep embeddings and positional encodings to create an embedding of the timestep vector. This embedding serves as a conditioning signal and is utilized to dynamically calculate the scaling and shifting parameters for AdaLN during the Transformer Layer Normalization step. This enables the normalization to incorporate the conditioning information of variable noise levels. We first consider the $l_a$ and $1 + l_v$ time-dimensions for audio and video embeddings respectively. When applying MoNL, we can easily keep track of the corresponding time segments among the $l_a$ and $1 + l_v$ dimensions, given the temporal compression factors in each modality. The noisy latents are then linearly projected matching the final dimension $d$ by adding appropriate spatiotemporal positional embeddings for video and temporal positional embeddings for audio, resulting in $d$ dimensional embeddings for each modality which are then concatenated.

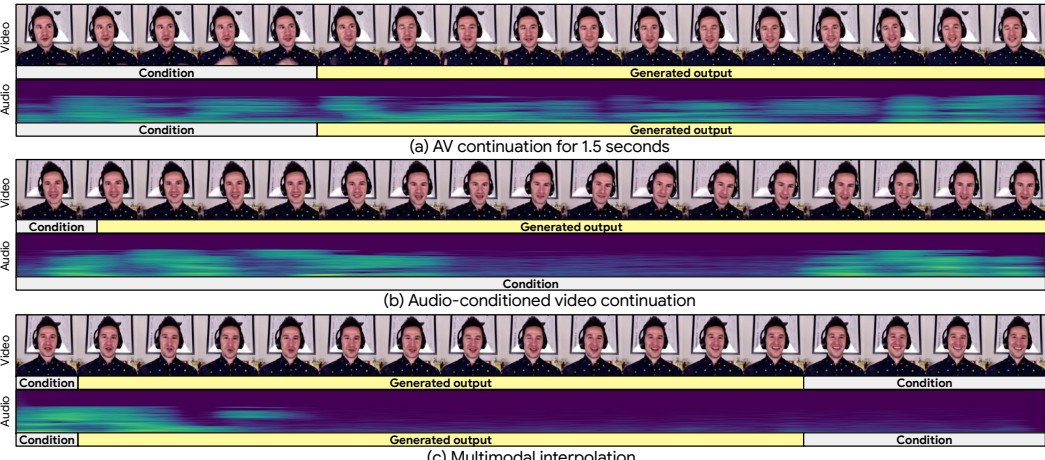

(a) AV continuation for 1.5 seconds

(b) Audio-conditioned video continuation

(c) Multimodal interpolation

Figure 6: Full length examples of our AVDiT trained with MoNL on the Monologue dataset. Samples were generated from unseen conditions at 8fps at $128\times128$ and are shown at the same rate.

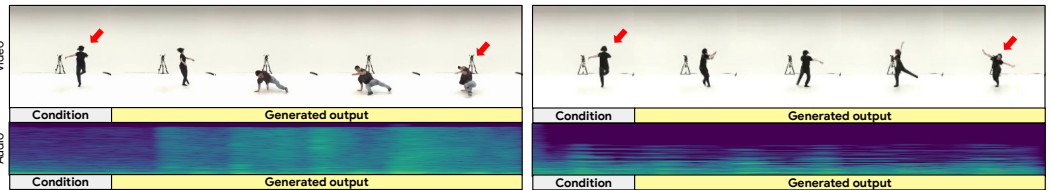

Figure 7: Unlike MM-Diffusion (left) where clothes and appearance is altered in the continuation (red arrow), our AVDiT with MoNL (right) maintains subject consistency in the AIST++ dataset.

## 5 Related Work

**Video diffusion models.** Diffusion models have revolutionized image [45, 35, 16, 39] and video generation with pixel-space [18, 17, 43] and latent-space [14, 56, 6, 9, 12] approaches. Recently, W.A.L.T [12] pushed the boundaries using transformer-based latent diffusion with joint image-video training. Tackling diverse audio-video generation tasks remains largely unexplored. With an AVDiT trained with MoNL, our unified approach empowers a single model to handle a range of tasks.

**Audio generative models.** Audio generation soared with WaveNet [36]'s autoregressive approach. Adversarial audio generation [27, 42, 26] emerged. Combining this with differentiable quantization [38, 47, 1] led to end-to-end neural codecs for efficient audio compression [21, 57]. Recently, diffusion models joined the fray, some using continuous latent spaces [29, 19, 11], others exploring discrete space [52]. Our AVDiT uses continuous embeddings from SoundStream for audio latents.

**Multimodal generative modeling.** While multimodal diffusion models [37, 41, 54, 39, 17, 23] have achieved impressive results, the field has primarily focused on the visual domain and audiovisual generation remains less explored. Existing approaches for audio-to-video [53, 53, 30] and video-to-audio [20, 33] generation typically learn task-specific conditional models, limiting their flexibility. To address this, recent works [40, 50] propose more versatile audiovisual models. However, they did not examine multimodal interpolation tasks, which we explore in this work. Tasks such as AV-continuation are critical to understand a model's capability to generate a temporally consistent multimodal sequence retaining the object consistency from the condition input.

## 6 Experiments

### 6.1 Datasets

**Monologues dataset** consists of 19.1 million videos for training and 25K videos for evaluation, each with a single person talking. The videos are center-cropped to a $256 \times 256$ resolution. This dataset includes a range of person appearances along with rich verbal and non-verbal communication cues. This dataset is ideally situated to assess concepts such as audiovisual gestural synchrony and multimodal expressions which are key components of human communication and interactions.

Table 1: Comparison of AVDiT trained with mixture of noise levels (MoNL) on the Monologues dataset for unconditional joint generation (`Joint`), cross-modal (`A2V`, `V2A`) and multimodal interpolation (`AV-inpaint`, `AV-continue`) tasks. FAD = 2.7 and FVD = 3.3 for groundtruth autoencoder reconstructions of the inputs. Fréchet metrics estimated with N=25k.

| Setting / Task | Joint | | A2V | V2A | AV-inpaint | | AV-continue | | Average | |
|---|---|---|---|---|---|---|---|---|---|---|
| | FAD ↓ | FVD ↓ | FVD ↓ | FAD ↓ | FAD ↓ | FVD ↓ | FAD ↓ | FVD ↓ | FAD ↓ | FVD ↓ |
| Conditional (task-specific) | 7.1 | **63.6** | 49.4 | 11.5 | 5.3 | 15.9 | 7.4 | 12.1 | 7.8 | 35.3 |
| Per modality | 7.0 | 84.4 | **34.1** | **4.7** | 6.2 | 213.6 | 4.5 | 92.1 | 5.6 | 106.1 |
| Vanilla | 7.1 | **63.6** | 53.3 | 8.1 | 8.1 | 226.8 | 6.1 | 140.8 | 7.4 | 121.1 |
| **MoNL (Ours)** | **6.4** | 77.6 | 40.2 | 5.3 | **4.6** | **11.8** | **3.1** | **8.8** | **4.9** | **34.6** |
| Ablations | | | | | | | | | | |
| Per time-segment | 6.6 | 96.3 | 124.5 | 12.1 | 5.1 | 28.2 | 5.0 | 72.3 | 7.2 | 80.3 |
| Per time-segment Per modality | 7.0 | 84.5 | 52.5 | 5.9 | 5.4 | 22.9 | 4.8 | 61.2 | 5.7 | 55.3 |
| `Pt/Pm/Ptm` | 9.0 | 90.1 | 43.1 | 5.1 | 5.2 | 13.4 | 4.1 | 16.9 | 5.9 | 40.9 |

Table 2: Quantitative comparison between our AVDiT with MoNL and MM-Diffusion (MMD).

| Task | Method | AIST++ | | | Landscape | | | |
|---|---|---|---|---|---|---|---|---|
| | | FAD ↓ | FVD ↓ | KVD ↓ | FAD ↓ | FVD ↓ | KVD ↓ | AV align ↑ |
| Reconstruction | | 0.90 | 11.72 | 0.96 | 0.76 | 16.41 | -0.25 | 0.60 |
| A2V | MMD | - | 184.45 | 33.91 | - | 238.33 | 15.14 | 0.54 |
| | Ours | - | **38.04** | **5.27** | - | **86.79** | **4.30** | **0.57** |
| V2A | MMD | 13.30 | - | - | 13.60 | - | - | 0.50 |
| | Ours | **1.11** | - | - | **0.78** | - | - | **0.51** |

Table 3: User study of comparison between our model and MM-Diffusion (MMD) on the AIST++ dataset.

| | Preference of ours over MMD | | |
|---|---|---|---|
| | AV align | AV quality | Person consistency |
| `AV-continue` | 0.69 | 0.71 | 0.93 |
| `A2V` | 0.77 | 0.61 | 0.75 |
| `V2A` | 0.61 | 0.49 | 0.60 |
| `Joint` | 0.74 | 0.72 | 0.81 |

**AIST++** is a subset of AIST [46] and contains 1,020 street dance videos (5.2 hours). The videos were segmented into in 8,233 samples for train and 110 for test at 10fps following Ruan et al. [40]. **Landscape** contains natural scenes from 928 [28] videos which were segemented into 5,400 samples for train and 600 samples for test at 10fps. We conduct most of the experiments on the Monologues due to its size and diversity. We use AIST++ and Landscape for comparison with MM-Diffusion.

## 6.2 Evaluation Settings

**Tasks.** We study three sets of tasks: (1) Joint audio-video (AV) generation (`Joint`): (2) Cross-modal generation: Audio-to-video (`A2V`) and Video-to-audio (`V2A`), and (3) AV interpolation generative tasks: `AV-inpaint` where a 1.5s clip is interpolated given one video frame, and 0.125s audio at the beginning and four video frames and 0.5s audio at the end, and `AV-continue` to fill out 1.5 seconds of AV given the first 5 video frames and corresponding 0.625s of audio.

**Baselines.** On the Monologues dataset, we compare the performance of AVDiT trained with MoNL versus three baselines: `Vanilla` (Eq. 2), `Conditional` models separately trained for each task, and the `per modality` model (Sec 3.2) which may be considered as a generalization of the UniDiffuser approach for sequences. We enabled the `Vanilla` model to generate cross-modal and multimodal interpolation outputs by using the replacement method [18, 40]. We also benchmark MoNL AVDiT against UNet-based MM-Diffusion (MMD) [40], the sole published work with a released model that tackles both audio and video generation within a single model. While a direct comparison between U-Nets and our transformer architecture is inherently challenging due to their distinct design principles, we show that MoNL AVDiT surpasses this strong U-Net baseline, demonstrating the effectiveness of the transformer architecture in this domain. We restrict our quantitative evaluation to A2V and V2A tasks because MMD fails to generate temporally consistent sequences in case of continuation tasks (see Figs. 2 and 7 for example).

**Quantitative evaluation.** We use Fréchet video distance (FVD) as our video evaluation metric following Yu et al. [55]. Similarly, we use Fréchet audio distance (FAD) as the audio evaluation metric following Ruan et al. [40]. Because we use latent space representations for the video and audio, we also report the FVD and FAD scores between reconstructed signal and the original signal as "ground-truth" scores as the performance upper bound. While we preferred user studies for assessing audio-video alignment as existing metrics miss subtle synchrony like dance moves matching music beats or gestures aligning with speech patterns, we computed AV-align score [33], limiting them to the open-domain Landscape dataset for the comparison with MMD.

**User studies.** We conducted user studies to evaluate the quality of generated content. We adopt the two axes of measurement introduced by Ruan et al. [40] namely audio/video quality and audio-video alignment, and introduce a third one, "subject consistency" to assess whether the person in the generated content is plausibly consistent with the input. For stimuli, we used a total of 360 generated

samples (not cherry picked) balanced across A2V, V2A, AV-continue and AV-inpaint tasks and for AVDiT trained with three approaches: MoNL, Vanilla and Per modality on Monologues dataset. The tasks were assessed on a 5-point Likert scale. We also compared rater preference for MoNL AVDiT vs. MMD with 30 videos and 5 raters per video. Raters were presented with generations from the two methods randomized as two options, A/B and were asked to pick one option for each of the three dimensions instead of using a 5-point scale. See more details on implementation and experimental setup in Secs. B and C in the Appendix.

## 6.3 Results

**Qualitative results.** As displayed in Figs. 1 and 6, Our AVDiT model trained with MoNL achieves impressive performance on various tasks within a single framework, including audio-to-video, video-to-audio, joint generation, multimodal continuation and interpolation with flexible input settings, generating temporally consistent videos. Notably, ours preserves clothing and appearance attributes during continuation tasks, unlike MMD which can alter these (see Figs. 2 and 7). More qualitative results and comparisons are available in Figs. 13, 14 and 15, and at avdit2024.github.io.

**Quantitative results.** As shown in Table 1, on average across all tasks, AVDiT trained with MoNL outperforms all baselines, demonstrating its versatility to learn diverse conditional distributions in a task-agnostic manner. MoNL excelled at generating samples that are temporally and perceptually consistent with the conditioning input, in the case of AV-inpaint and AV-continue tasks, where other baselines generally failed. Per-modality approach surpassed MoNL for A2V and V2A tasks consistent with the findings in Bao et al. [3] likely because conditional distributions in these cases only need to capture cross-modal associations and not necessarily the underlying temporal dynamics. Unsurprisingly, the vanilla diffusion model trained for joint generation exhibited superior performance in this specific scenario but served as a lower-bound of performance for all other tasks. Finally, MoNL performed better than (if not on-par with) task-specific models for all conditional tasks.

As evident from Table 2, MoNL outperformed MMD in terms of the FAD and FVD metrics across all tasks on the AIST++ and Landscape datasets, as estimated using the code provided by Ruan et al. [40]. The significantly better audio generation in our model, likely due to the combination of MoNL and our choice of the SoundStream audio autoencoder, is also reflected in the ground-truth FAD scores for audio reconstruction. In case of video reconstruction quality, (ground-truth FVD) on AIST++, our choice of autoencoder was inferior to MMD, possibly due to the small dataset size. Qualitatively, we observed that the MAGVIT-v2 reconstructions eliminated flickering across frames but the reconstruction of small face regions in AIST++ dance videos was blurry. These findings should be interpreted cautiously due to several factors: the limited size of the AIST++ and Landscape training splits, our use of a transformer backbone versus MMD's coupled U-Nets, and our use of pretrained autoencoders for latent space representations. On the Landscape dataset, AV-align results demonstrate that our model achieves better alignment compared to MMD, which aligns with the findings from the user study below.

**User studies.** A comparison of the distribution of Likert scores across all tasks for the three approaches we compared is shown in Fig. 8. Pairwise Mann-Whitney $U$ tests were conducted with Bonferroni correction for multiple comparisons to assess statistical difference. Across all axes, raters preferred samples generated from MoNL over that of Vanilla or Per-modality (Pm) approaches. Examining task-specific trends (see Fig. 12 in the Appendix), for the cross-modal tasks, Pm was rated significantly higher than Vanilla, and there was no significant difference between MoNL and Pm (except for the V2A task on AV alignment). For multimodal interpolation tasks, MoNL was rated significantly higher than Pm. In line with quantitative results, these results suggest that MoNL excelled at generating samples, that are perceptually and temporally consistent with the input conditioning.

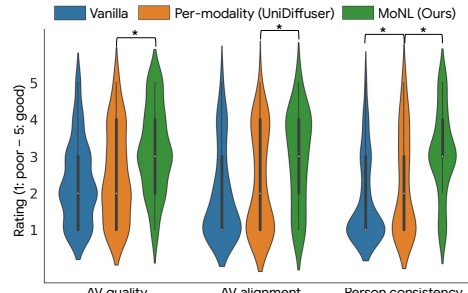

Figure 8: Comparative analysis across AVDiT models from the user study on AV quality, AV alignment and person consistency. The * indicates statistically significant pairwise difference at $p < 0.01$ after multiple correction.

As indicated in Table 3, our MoNL AVDiT outperformed MMD in user studies, especially in consistency where ours showed improved consistency along factors such as person's appearance

or the attire. MMD was preferred slightly more in V2A tasks, possibly because the Soundstream audtoencoder we used for audio was not optimized for music generation like in MMD.

**Ablations.** Recall that MoNL training randomly selects one of the four timestep designs described in Sec. 3.2. We compare MoNL with Per time-segment `Pt`, and Per time-segment and Per modality `Ptm` and `Pt/Pm/Ptm` that excludes vanilla from the timestep mixture, approaches separately as shown in Table 1. Overall, `Ptm` noise excelled at inpainting and continuation tasks though it was not on par with per-modality approach for cross-modal tasks. In general, `Pt` does not perform well by itself. In our experiments, we also observed that the combination of `Pm`, `Pt`, `Ptm` and `Pt/Pm/Ptm` was sufficient for comparable performance on most tasks except for unconditional joint generation. Adding the `Vanilla` approach to the mixture of timesteps improved performance for unconditional joint generation while not substantially compromising the performance on other tasks.

# 7   Conclusion

We propose a unified approach for multimodal diffusion using a mixture of noise levels (**MoNL**) for generating and manipulating sequences across modalities and time. This empowers a single model to handle diverse tasks like audio-video continuation, interpolation, and cross-modal generation. We show that an audiovisual latent diffusion transformer (**AVDiT**) trained with MoNL achieves state-of-the-art performance in audiovisual-sequence generation, providing new opportunities for expressive and controllable multimedia content creation.

See Sec. A in the Appendix for discussions on limitations and considerations.

## Acknowledgements

We thank Kihyuk Sohn, Caroline Pantofaru and Brian Eoff for feedback on early versions of the manuscript. We thank Prof. Se Young Chun for helpful discussions and feedback on the theoretical aspects of this work. Special thanks to Alex Siegman and Xuhui Jia for managing compute resources.

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

# Appendix

## A    Limitations, Impact and Considerations

Our proposed approach, combining mixture of noise levels (MoNL) with the generative capabilities of the Audiovisual diffusion transformer (AVDiT), has certain limitations. As shown with demo videos on avdit2024.github.io, while our models effectively capture subject consistency and intricate nonverbal behaviors such as gestural synchrony with vocal tone, significant improvements are necessary to enhance visual and speech quality. Future research will concentrate on super-resolution systems to address visual quality, while text conditioning could potentially further optimize speech quality.

A key focus in this work was to demonstrate the versatility and use of MoNL across various tasks using a simple mixing scheme by randomly choosing between different timestep candidates as representative schemes for applying variable noise levels. This simplicity showcases its broad applicability. We acknowledge that fine-grained controlling by weighted mixing of the different schemes could be explored for specific goals or tasks in future work. In fact, one can imagine an arbitrarily large number of timestep candidates in the vector space of the inputs. We specifically chose a simple mixture scheme to demonstrate its versatility as a proof of concept, rather than optimize for any single task.

Although the method presented in this work is for general multimodal applications, our experiments included human-centric generation tasks. This enabled us to explore unique challenges of that problem setting. For example, consider the case of perceptual expectations for audiovisual alignment/coherence, where misaligned audio and visual cues can drastically alter perception of speech [34, 24]. Generation of photo-realistic persons, speech, and joint generation of both can perpetuate stereotypes. We recognize the ethical concerns and underscore that our goal here is to explore how understanding aspects such as nonverbal behavior in multimodal communications using generative models can open up new avenues in research.

Specifically, the A2V and V2A tasks in human-centric context, which involve extrapolating visual appearance from speech and vice versa, have the potential to perpetuate stereotypes. The generated samples are derived from the model's understanding of cross-modal associations in the training dataset, which can be vastly different from human perception. One possible mitigation is to ensure that the model can generate diverse outputs for a given input. Diffusion models can achieve this by utilizing different noises at inference time, given a sufficiently large and diverse training dataset. A recent study showed that Diffusion models demonstrate better sample diversity in generations compared to GANs [4], however, addressing potential issues around mode collapse in the generative models, especially with multimodal data is an open research problem.

Our work also introduced "consistency" to qualitatively assess whether the generation remains congruent with the input conditioning. In continuation and interpolation tasks, the disparity between what the model generates and human perception can be generally minimal, as the conditioning provides a perceptual template for the subject's potential appearance or voice. In contrast, A2V and V2A tasks warrant an in-depth analysis of this disparity. As our immediate goal was to assess the capability of our proposed approach (and baselines) to generate samples from various conditional distributions, we focused on a broad definition for measuring consistency in user studies. Our future work will focus on extending the consistency measure for a granular understanding of these biases by (1) comparing cross-modal associations in the training data to that of the generated samples, and (2) disaggregate model and human evaluations in cross-modal generation tasks by identifying specific dimensions of human appearance attributes like perceived gender expression or human communication aspects such as voice and gestural synchrony using diverse rater pools.

## B    Implementation Details

**Autoencoders and AVDiT.** Given the domain specific nature of the datasets. we trained dataset-specific MAGVIT-v2 autoencoders following Yu et al. [55]. For the Monologues dataset, we downsampled the data to 8fps and $128 \times 128$ resolution for video and 16kHz for audio and randomly

sampled a contiguous clip of 2.125 second (17 frames) to match the input requirements of MAGVIT-v2. This resulted in a dataset of about 11.8K hours for training. The spatial and temporal video compression ratios were set to $r_s = 8$ and $r_{t_v} = 4$, whereas the temporal audio compression ratio was $r_{t_a} = 320$. The embedding dimension of the video and audio latent spaces are $d_v = 8$ and $d_a = 1024$ respectively, with the target embedding dimension after linear projection, $d = 1024$. All latents were zero-mean and unit-variance normalized with empirical mean and variance estimates on a small subset. AVDiT has 24 transformer layers with 16 heads with MSA with a total of 420M parameters.

**Diffusion training and inference.** During training, we use a linear noise variance schedule and a diffusion step $T = 1000$, and a self-conditioning rate of 0.9 following Gupta et al. [12]. At the inference time, we use 250 DDIM steps. All models were trained for about 400K steps with a batch size of 256. We used the AdamW optimizer [31] with a learning rate of 5e-4, 5K warm-up steps, cosine learning rate scheduler and EMA consistent with the denoising transformer setting in Gupta et al. [12].

**Compute resources.** Each experiment listed in Table 1 was conducted using 256 v5e TPU chips (with $16 \times 16$ topology) for training (on average, the models were trained for around 350K steps with a batch size of 256 for around five days); inference was conducted using 16 v5e TPU chips with a topology of $4 \times 4$. See https://cloud.google.com/tpu/docs/v5e for more details. Benchmarking experiments to compare with MM-Diffusion on the AIST++ and Landscape datasets were conducted using two A100 GPUs for conditional inference and estimating FAD/FVD metrics.

# C  Experimental Details

## C.1  Evaluation metrics

Since our primary use case is speech generation with the Monologues dataset, we use VGGish embeddings as feature for FAD estimation [22] for the results reported in Table 1. For AV interpolation generative tasks (AV-continue and AV-inpaint), we carefully excluded the conditioning AV frames while estimating Fréchet metrics.

## C.2  Comparison with MM-Diffusion

In order to conduct a fair comparsion to the results reported in MM-Diffusion publication [40], we use the data preprocessing and evaluation code provided at github.com/researchmm/MM-Diffusion for FVD, FAD and KVD metrics. Note that the FAD computation here does not use VGGish embeddings. Instead, it uses AudioCLIP [13] which was trained for general sound classification tasks and not suitable for speech generation tasks as in the Monologues dataset reported in Table 1.

For the FAD, FVD and KVD results reported in Table 2, we match training conditions for the input image resolution and video FPS with Ruan et al. [40], i.e., $64 \times 64$ resolution images at 10fps. We match the duration of audio-video from both models to 2 seconds. For visualization purpose in Fig. 2, we also train our models with $256 \times 256$ resolution to match the super-resolution output resolution used by MM-Diffusion.

Ruan et al. [40] introduce a method for implementing zero-shot transfer of A2V and V2A tasks, inspired by the reconstruction-guided sampling proposed by Ho et al. [18]. For instance, in V2A tasks, the generated noisy audio $\tilde{a}_t$ is computed at each step as follows:

$$a_t, v_t = \theta_{av}(a_{t+1}, \hat{v}_{t+1}), \tag{8}$$

$$\tilde{a}_t = a_t - \lambda\sqrt{1 - \overline{\alpha}_t}\nabla_{a_t}||v_t - \hat{v}_t||_2^2 \tag{9}$$

where $a_{t+1}, \hat{v}_{t+1}$ are a $N$-length sequence of generated noisy audio and conditioned noisy video at $t + 1$, $\theta_{av}$ is a parameterized denoising step, and $\lambda$ is a gradient weight. Similarly, the zero-shot transfer of AV-continuation task using the reconstruction-guided sampling [18] can be described by the following equations:

$$a_t, v_t = \theta_{av}(a_{t+1}, v_{t+1}), \tag{10}$$

$$\tilde{a}_t^{(n_c+1:N)} = a_t^{(n_c+1:N)} - \lambda\sqrt{1 - \overline{\alpha}_t}\nabla_{a_t}\left(\left\|v_t^{(1:n_c)} - \hat{v}_t^{(1:n_c)}\right\|_2^2 + \left\|a_t^{(1:n_c)} - \hat{a}_t^{(1:n_c)}\right\|_2^2\right) \tag{11}$$

$$\tilde{v}_t^{(n_c+1:N)} = v_t^{(n_c+1:N)} - \lambda\sqrt{1 - \overline{\alpha}_t}\nabla_{v_t}\left(\left\|v_t^{(1:n_c)} - \hat{v}_t^{(1:n_c)}\right\|_2^2 + \left\|a_t^{(1:n_c)} - \hat{a}_t^{(1:n_c)}\right\|_2^2\right), \tag{12}$$

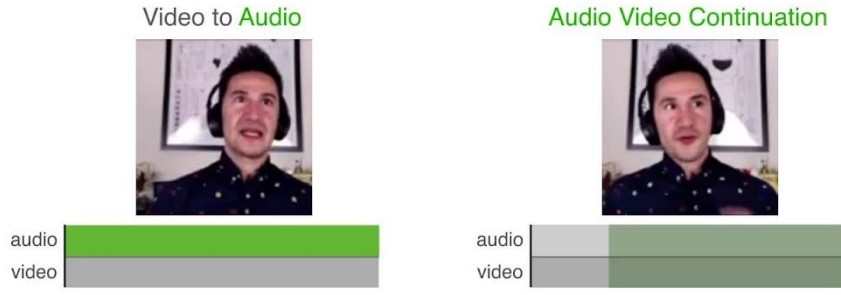

Figure 9: Example stimuli shown to the raters for the user study. We conducted user studies for four tasks, A2V, V2A, audiovisual continuation and multimodal interpolation tasks. One track each for the audio and video modality below the stimulus video were shown to effectively convey the portions that were generated (in green) and condition input (gray).

Table 4: Rater instructions for audio/video quality metric.

| Score | Audio / Video quality |
|-------|----------------------|
| 1 | Pure noise, completely UNRECOGNIZABLE CONTENT |
| 2 | The generated content has natural structure in SOME places, but not most |
| 3 | The generated content has natural structure in MOST places |
| 4 | The generated content is NATURAL, BUT can be recognized as GENERATED content |
| 5 | The generated content is so NATURAL that it is indistinguishable from the REAL-WORLD |

Table 5: Rater instructions for audio-video alignment metric.

| Score | Audio-video alignment |
|-------|----------------------|
| 1 | The audio-video are total noise and are completely IRRELEVANT |
| 2 | The generated content has CORRELATION between audio and video in SOME segments, but not most |
| 3 | The generated content has CORRELATION between audio and video in MOST segments |
| 4 | The generated content has NATURAL CORRELATION between the audio and video, BUT can be recognized as GENERATED content |
| 5 | The generated content has CORRELATION between the audio and video indistinguishable from the REAL-WORLD |

Table 6: Rater instructions for subject consistency.

| Score | Subject consistency |
|-------|---------------------|
| 1 | The person generated is INCONSISTENT with the input |
| 2 | The person generated is CONSISTENT with content in SOME SEGMENTS |
| 3 | The person generated is CONSISTENT with the generated content in MOST SEGMENTS |
| 4 | The person generated is NATURALISTIC, but can be recognized as GENERATED content |
| 5 | The person generated is INDISTINGUISHABLE from the REAL-WORLD |

where $n_c$ is the number of conditioned time-segments. We closely follows their V2A codebase to faithfully execute the continuation task. We adopt $\lambda = 0.02$ to prevent numerical instability, as the results tend to diverge for $\lambda > 0.02$.

## C.3 Qualitative Evaluation

Examples of the video stimulus template shown to the raters is presented in Fig. 9. The rater instructions provided for each axis of quality, alignment and consistency are shown in Tables 4, 5 and 6 respectively.

---

**Algorithm 1** Sampling of a diffusion timestep vector

---

1: **function** GETTIMESTEPVEC(type)
2:     **if** type == MoNL **then**
3:       type $\sim \mathcal{U}(\{\texttt{Vanilla}, \texttt{Pt}, \texttt{Pm}, \texttt{Ptm}\})$
4:     $\boldsymbol{t}_{ref} \in \mathbb{R}^{M \times N} \sim \mathcal{U}(\{1, 2, \ldots, T\})$
5:     $\boldsymbol{t} = \boldsymbol{0} \in \mathbb{R}^{M \times N}$
6:     **for** $m = 1, \ldots, M$ **do**
7:       **for** $n = 1, \ldots, N$ **do**
8:         **if** type == Vanilla **then**
9:           $t^{(m,n)} = t_{ref}^{(1,1)}$
10:         **else if** type == Pt **then**
11:           $t^{(m,n)} = t_{ref}^{(1,n)}$
12:         **else if** type == Pm **then**
13:           $t^{(m,n)} = t_{ref}^{(m,1)}$
14:         **else if** type == Ptm **then**
15:           $t^{(m,n)} = t_{ref}^{(m,n)}$
16:       **end for**
17:     **end for**
18:     **return** $\boldsymbol{t}$
19: **end function**

---

**Algorithm 2** Training with MoNL

---

    **input** $q(\boldsymbol{z}_0), \boldsymbol{\epsilon_\theta}, \texttt{type}$ (timestep sample type)
1: **repeat**
2:   $\boldsymbol{z}_0 \sim q(\boldsymbol{z}_0)$
3:   $\boldsymbol{\epsilon} \sim \mathcal{N}(\boldsymbol{0}, \boldsymbol{I})$
4:   $\boldsymbol{t} = \text{GETTIMESTEPVEC}(\texttt{type})$
5:   **for** $m = 1, \ldots, M$ **do**
6:     **for** $n = 1, \ldots, N$ **do**
7:       $z_{t(m,n)}^{(m,n)} = \sqrt{\overline{\alpha}_{t^{(m,n)}}} z_0^{(m,n)} + \sqrt{1 - \overline{\alpha}_{t^{(m,n)}}} \epsilon^{(m,n)}$
8:     **end for**
9:   **end for**
10:   Take gradient step on
11:   $\nabla_{\boldsymbol{\theta}} \|\boldsymbol{\epsilon_\theta}([z_{t(1,1)}^{(1,1)}, \ldots, z_{t(M,N)}^{(M,N)}], \boldsymbol{t}) - \boldsymbol{\epsilon}\|_2^2$
12: **until** converged

---

**Algorithm 3** Joint generation of $\boldsymbol{z}_0$

---

1:  $\hat{\boldsymbol{z}}_T \in \mathbb{R}^{M \times N \times d} \sim \mathcal{N}(\boldsymbol{0}, \boldsymbol{I})$
2:  **for** $\tau = T, \ldots, 1$ **do**
3:    $\boldsymbol{\epsilon} \in \mathbb{R}^{M \times N \times d} \sim \mathcal{N}(\boldsymbol{0}, \boldsymbol{I})$ if $\tau > 1$, else $\boldsymbol{\epsilon} = \boldsymbol{0}$
4:    $\boldsymbol{t} \in \mathbb{R}^{M \times N} = \tau \boldsymbol{I}$
5:    $\hat{\boldsymbol{z}}_{\tau-1} = \frac{1}{\sqrt{\alpha_\tau}} \left( \hat{\boldsymbol{z}}_\tau - \frac{\beta_\tau}{\sqrt{1 - \overline{\alpha}_t}} \boldsymbol{\epsilon_\theta}(\hat{\boldsymbol{z}}_\tau, \boldsymbol{t}) \right) + \sigma_\tau \boldsymbol{\epsilon}$
6:  **end for**
7:  **return** $\hat{\boldsymbol{z}}_0$

---

**Algorithm 4** Cross-modal generation of $\hat{\boldsymbol{z}}_0 \in \mathbb{R}^{(M-m_c) \times N \times d}$ conditioned on $\boldsymbol{z}_0 \in \mathbb{R}^{m_c \times N \times d}$

---

1:  $\hat{\boldsymbol{z}}_T \in \mathbb{R}^{(M-m_c) \times N \times d} \sim \mathcal{N}(\boldsymbol{0}, \boldsymbol{I})$
2:  $\boldsymbol{t} \in \mathbb{R}^{M \times N} = \boldsymbol{0}$
3:  **for** $\tau = T, \ldots, 1$ **do**
4:    $\boldsymbol{\epsilon} \in \mathbb{R}^{(M-m_c) \times N \times d} \sim \mathcal{N}(\boldsymbol{0}, \boldsymbol{I})$ if $\tau > 1$, else $\boldsymbol{\epsilon} = \boldsymbol{0}$
5:    $\boldsymbol{t}^{(m_c+1:M,1:N)} = \tau \boldsymbol{I}$
6:    $\hat{\boldsymbol{\epsilon}} = \boldsymbol{\epsilon_\theta}^{(m_c+1:M,N)}([\boldsymbol{z}_0, \hat{\boldsymbol{z}}_\tau], \boldsymbol{t})$
7:    $\hat{\boldsymbol{z}}_{\tau-1} = \frac{1}{\sqrt{\alpha_\tau}} (\hat{\boldsymbol{z}}_\tau - \frac{\beta_\tau}{\sqrt{1 - \overline{\alpha}_\tau}} \hat{\boldsymbol{\epsilon}}) + \sigma_\tau \boldsymbol{\epsilon}$
8:  **end for**
9:  **return** $\hat{\boldsymbol{z}}_0$

---

## D  Algorithms

The core algorithms for implementing mixture of noise levels (**MoNL**) are presented in three parts: sampling of diffusion timestep vector (Algorithm 1), training process (Algorithm 2), and joint/cross-modal generation at inference time (Algorithms 3 and 4). The sampling algorithms are flexible, using DDPM [16] as an example, and can be replaced with other efficient learning-free samplers like DDIM [45] or Analytic-DPM [2]. Notably, conditional generation across time-segments is simply a change of axis in Algorithm 4.

## E  Gratis Classifier-Free Guidance

**Classifier-free guidance** (CFG) [15], a technique designed to enhance the quality of samples produced by conditional diffusion models using a linear combination of the conditional and unconditional outputs as follows:

$$\hat{\epsilon}_{\boldsymbol{\theta}}(\boldsymbol{x}_t, \boldsymbol{y}_0, t) = (1+s)\epsilon_{\boldsymbol{\theta}}(\boldsymbol{x}_t, \boldsymbol{y}_0, t) - s\epsilon_{\boldsymbol{\theta}}(\boldsymbol{x}_t, t) \tag{13}$$

$$= (1+s)\epsilon_{\boldsymbol{\theta}}^{\text{cond}} - s\epsilon_{\boldsymbol{\theta}}^{\text{uncond}} \tag{14}$$

where $s$ is a guidance scale and the conditional and unconditional outputs are denoted by $\epsilon_{\boldsymbol{\theta}}^{\text{cond}}$ and $\epsilon_{\boldsymbol{\theta}}^{\text{uncond}}$ respectively. Typically, a null token $\varnothing$, is used to allow the conditional model to generate unconditional outputs by setting $\boldsymbol{y}_0 = \varnothing$.

**Gratis CFG.** CFG (Eq. 13) is supported in our framework at inference without any additional training similar to UniDiffuser [3]. Instead of using a null token for generating unconditional outputs ($\epsilon_{\boldsymbol{\theta}}^{\text{uncond}}$ in Eq. 13), Gaussian noise is injected to the conditional portions input per task specification, and setting $t^{(m,n)} = T$. Conditional outputs $\epsilon_{\boldsymbol{\theta}}^{\text{cond}}$ are obtained as illustrated in Fig. 4. Our vector formulation of the timestep allows us to apply varying levels of noise to different parts of the input. This opens up a number of possibilities for constructing various CFG forms by emphasizing different time segments or modalities, depending on the task at hand.

MoNL supports classifier-free guidance (CFG) without requiring additional design. Unlike the original CFG (see Eq. 13), it does not need a null token either, hence *gratis* or free. This is achieved by injecting Gaussian noise to the conditional portions of the multimodal space and setting $t^{(m,n)} = T$ for the output as illustrated in Fig. 10 for the case of cross-modal generation of audio-in, video-out. To illustrate mCFG, consider the conditional output of the network in the cross-modal task (see Eq. 7), denote term used in the gradient step as $\mathbf{Z}_{\boldsymbol{t}}^{(1:M,1:N)} = [z_{t^{(1,1)}}^{(1,1)}, \ldots, z_{t^{(M,N)}}^{(M,N)}]$ and conditional portions as $\epsilon_{\boldsymbol{\theta}}^{\text{cond}} = \epsilon_{\boldsymbol{\theta}}(\mathbf{Z}_{\boldsymbol{t}}^{(1:M,1:N)}, \boldsymbol{t})$ where

$$\mathbf{Z}_{\boldsymbol{t}}^{(1:m_c,1:N)} = \boldsymbol{z}_0^{(1:m_c,1:N)}, \quad \mathbf{Z}_{\boldsymbol{t}}^{(m_c+1:M,1:N)} = \boldsymbol{z}_t^{(m_c+1:M,1:N)}$$

$$\boldsymbol{t}^{(1:m_c,1:N)} = 0, \qquad \boldsymbol{t}^{(m_c+1:M,1:N)} = t.$$

Then, the output for the cross-modal generation task is:

$$\epsilon_{\boldsymbol{\theta}}^{\text{uncond}} = \epsilon_{\boldsymbol{\theta}}(\mathbf{Z}_{\boldsymbol{t}}^{(1:M,1:N)}, \boldsymbol{t}). \tag{15}$$

where

$$\mathbf{Z}_{\boldsymbol{t}}^{(1:m_c,1:N)} = \boldsymbol{z}_0^{(1:m_c,1:N)}, \quad \mathbf{Z}_{\boldsymbol{t}}^{(m_c+1:M,1:N)} = \boldsymbol{z}_T^{(m_c+1:M,1:N)}$$

$$\boldsymbol{t}^{(1:m_c,1:N)} = 0, \qquad \boldsymbol{t}^{(m_c+1:M,1:N)} = T.$$

where $\boldsymbol{h}_T \sim \mathcal{N}(\boldsymbol{0}, \boldsymbol{I})$. Then, mCFG operates by blending the conditional $\epsilon_{\boldsymbol{\theta}}^{\text{cond}}$ and unconditional portions $\epsilon_{\boldsymbol{\theta}}^{\text{uncond}}$ per task specification as follows:

$$\hat{\epsilon}_{\boldsymbol{\theta}} = (1+s)\epsilon_{\boldsymbol{\theta}}^{\text{cond}} - s\epsilon_{\boldsymbol{\theta}}^{\text{uncond}}. \tag{16}$$

where $s$ is a guidance scale.

$$\hat{\epsilon}_{\boldsymbol{\theta}}^{(1:M,n_c+1:N)} = (1+s)\epsilon_{\boldsymbol{\theta}}^{\text{cond},(1:M,n_c+1:N)} - s\epsilon_{\boldsymbol{\theta}}^{\text{uncond},(1:M,n_c+1:N)} \tag{17}$$

By formulating the timestep as a vector, we can apply varying levels of noise to different input components. This unlocks diverse possibilities for crafting varied CFG structures. Each structure

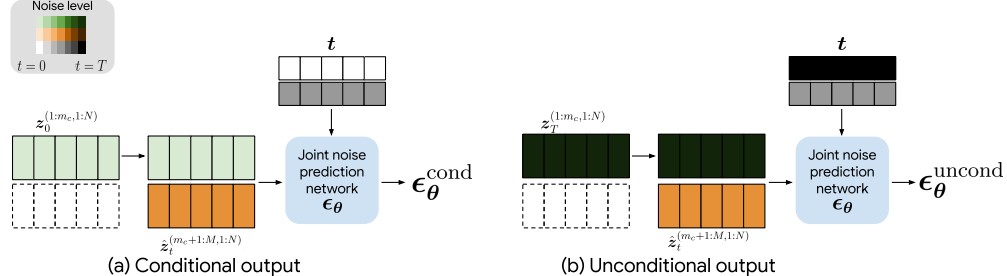

(a) Conditional output        (b) Unconditional output

Figure 10: Application of CFG for free in our MoNL approach for cross-modal generation tasks. Whereas a null token is used in traditional CFG for unconditional output, formulating diffusion timestep as a vector enables this by setting the input condition per task-specification to pure noise.

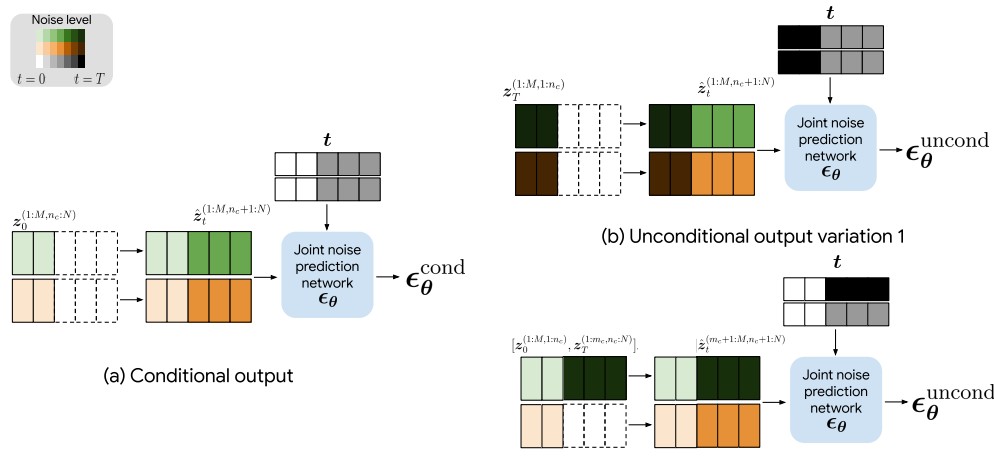

(a) Conditional output

(b) Unconditional output variation 1

(c) Unconditional output variation 2

Figure 11: Application of CFG for free in our MoNL approach for multimodal interpolation tasks. Because our vector formulation of the timestep enables applying variable noise levels to different portions of the input one can construct a different CFG with "mix-and-match" of modalities and time-segments for creating unconditional outputs. Here, we show an example for multimodal interpolation task for (a) conditional output with two variations: (b) unconditional output with respect to input condition per task specification, (c) partial conditional output, but unconditional output with respect to modalities.

can amplify specific time segments or modalities based on the task demands, as demonstrated in Fig. 10 for cross-modal tasks and Fig. 11 for the case of multimodal interpolation generation driven by temporal conditioning.

# F    Discussion

**Limited conditioning information:** In Table 7, we compares the results of AV continuation depending on the input information: `AV-continue-2s`) to fill out 2 seconds of AV given the first video frame and corresponding 0.125s of audio `AV-continue-1.5s`) to fill out 1.5 seconds of AV given the first 5 video frames and corresponding 0.625s of audio. The model performed better when given more context (5 video frames and corresponding audio) compared to less context (1 frame and corresponding audio), even though task-specific training can be an upper bound for performance. This suggests that limited conditioning information can lead to issues like unnatural motion and inconsistencies, and including more context improves the model's performance.

Table 7: Comparison of AVDiT trained with mixture of noise levels (MoNL) on the Monologues dataset for `AV-continue-2s`) to fill out 2 seconds of AV given the first video frame and corresponding 0.125s of audio `AV-continue-1.5s`) to fill out 1.5 seconds of AV given the first 5 video frames and corresponding 0.625s of audio. FAD = 2.7 and FVD = 3.3 for ground truth autoencoder reconstructions of the inputs. Fréchet metrics estimated with N=25k.

| Setting/Task | AV-continue-2s | | AV-continue-1.5s | |
|---|---|---|---|---|
| | FAD↓ | FVD↓ | FAD↓ | FVD↓ |
| Conditional (task-specific) | 8.2 | 117.3 | 7.4 | 12.1 |
| Per modality | 5.8 | 120.5 | 4.5 | 92.1 |
| Vanilla | 7.5 | 142.6 | 6.1 | 140.8 |
| MoNL (Ours) | 3.6 | 12.9 | 3.1 | 8.8 |
| Per time-segment | 6.7 | 102.5 | 5 | 72.3 |
| Per time-segment Per modality | 5.8 | 82.8 | 4.8 | 61.2 |
| Pt/Pm/Ptm | 4.1 | 20.2 | 4.1 | 16.9 |

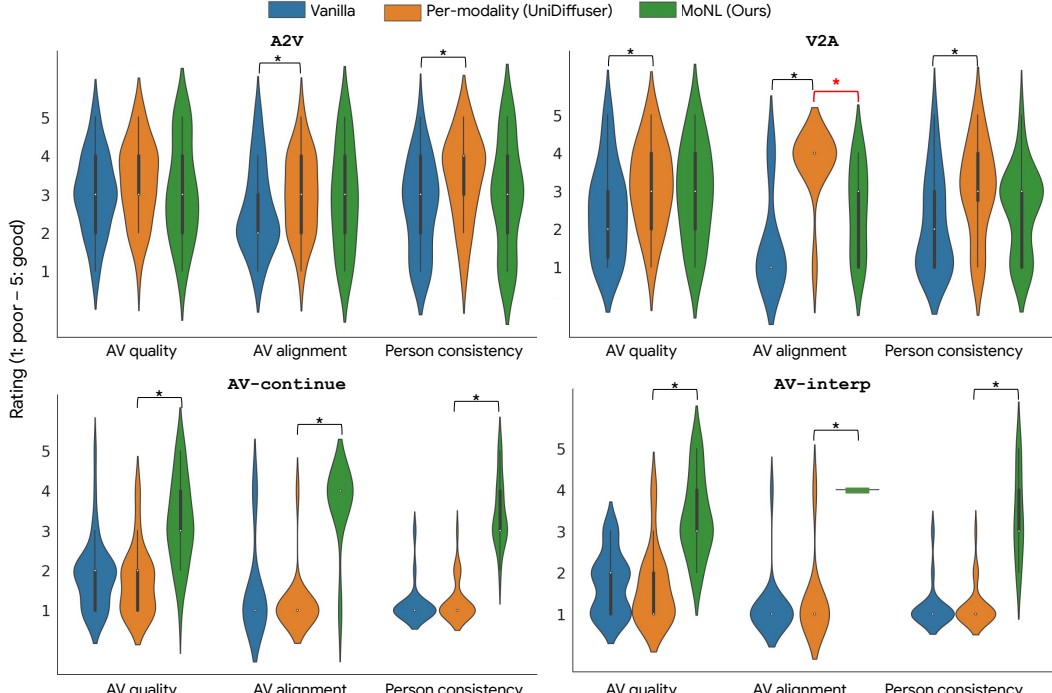

Figure 12: Comparative analysis across AVDiT models from the user study along axes of AV quality, AV alignment and person consistency for two cross-modal generation tasks (A2V and V2A), and multimodal interpolation tasks (`AV-continue` and `AV-inpaint`). The * indicates statistically significant pairwise difference at $p < 0.01$ after multiple comparison correction. Across the board, MoNL (Ours) was rated significantly better or on par across all tasks, except for AV-alignment for V2A task (comparison shown in red). For A2V task, there was no significant difference between the models compared for the measure of AV quality. For multimodal interpolation tasks (bottom row), Our approach far surpasses other models for quality, alignment and consistency underscoring the ability of our approach to generate temporally consistent samples that are perceptually congruent with the input condition.

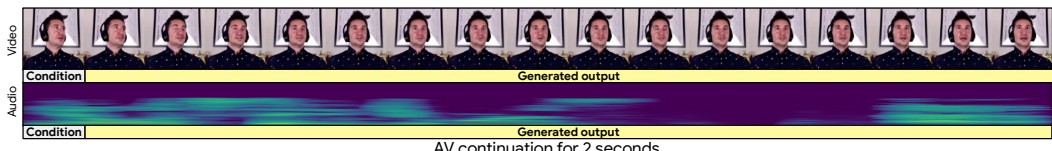

Figure 13: Full length examples of AV continuation for 2s from AVDiT trained with MoNL. Samples were generated at 8 fps at $128 \times 128$ resolution and are shown at the same rate.

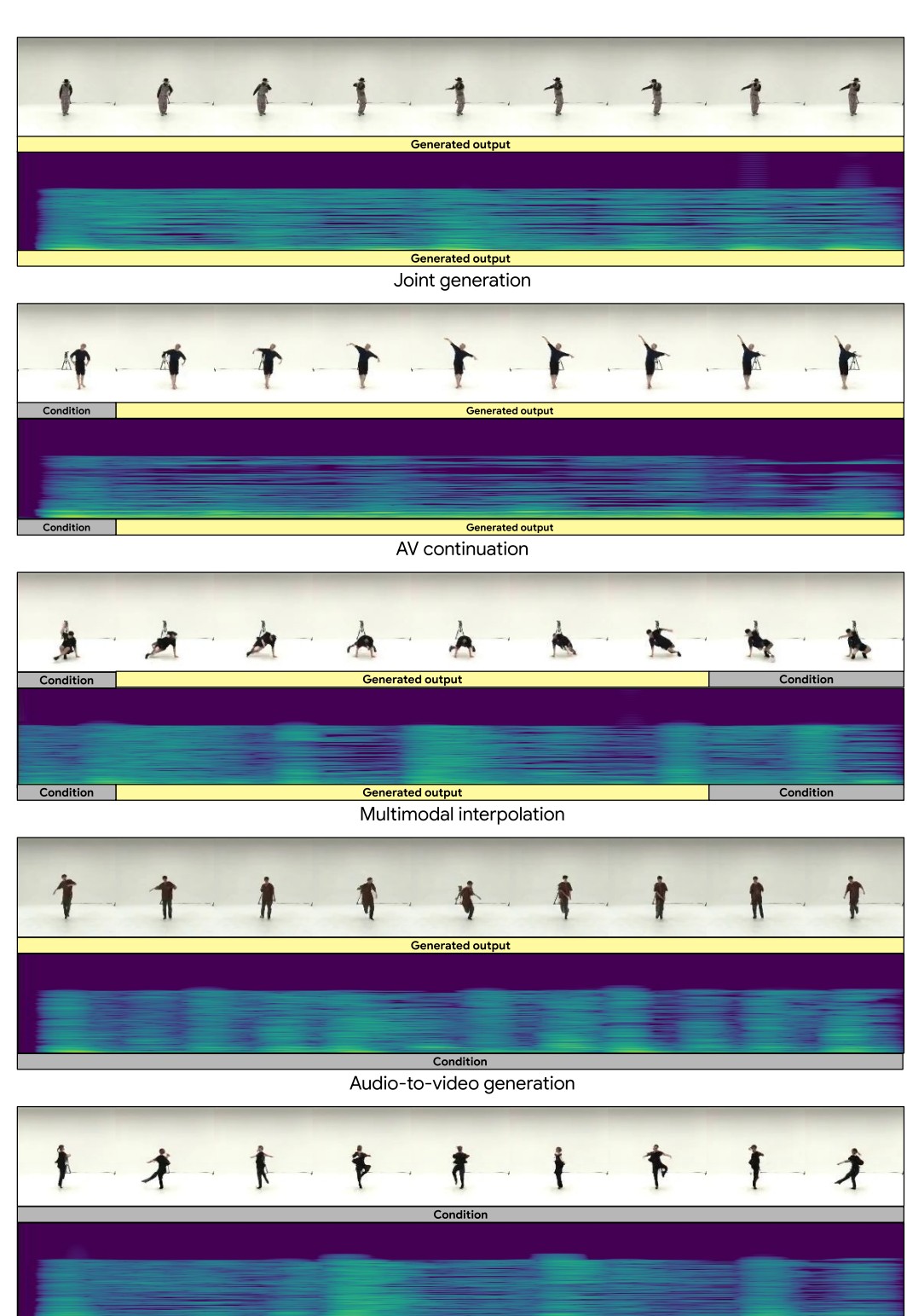

Figure 14: Full length examples of generations from AVDiT trained with mixture of noise levels (MoNL) on the AIST++ dataset. Generated at 8fps with $128 \times 128$ image resolution.

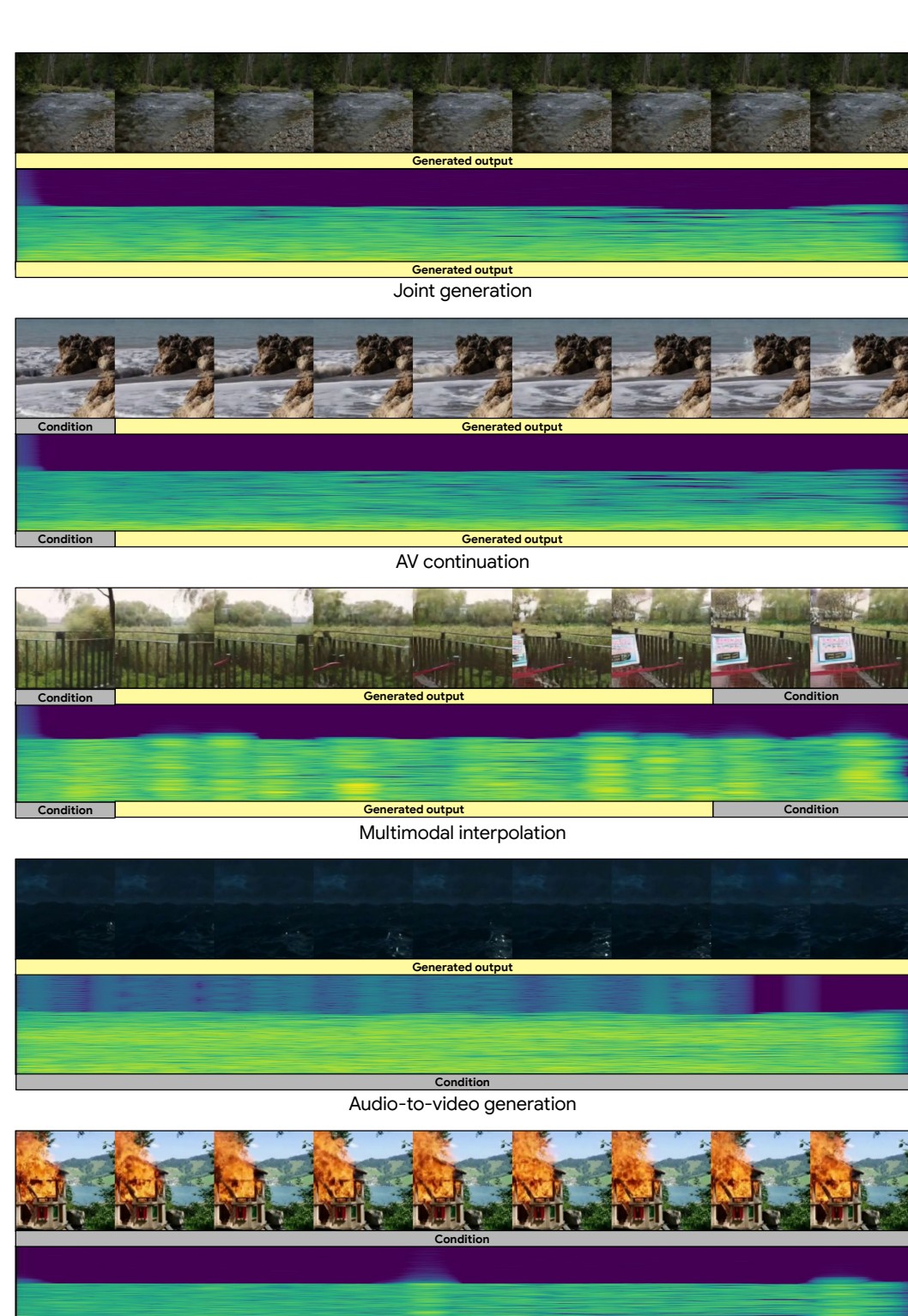

Figure 15: Full length examples of generations from AVDiT trained with mixture of noise levels (MoNL) on the Landscape dataset. Generated at 8fps with $256{\times}256$ image resolution.

# G Theoretical Background on Mixture of Noise Levels

## G.1 Theoretical Background on Multimodal Learning

In "A Theory of Multimodal Learning" [32], multimodal learning is shown to offer a superior generalization bound compared to unimodal learning, with an improvement factor of $O(\sqrt{n})$, where $n$ denotes the sample size. This benefit relies on connection and heterogeneity between modalities:

**Connection.** The bound depends on learned connections between $(\mathcal{X})$ and $(\mathcal{Y})$.

**Heterogeneity.** Describes how modalities $\mathcal{X}$ and $\mathcal{Y}$, diverge and complement.

If connection and heterogeneity are missing, ill-conditioned scenarios can arise. For instance, if $x = y$, perfect connection suggests no need for learning about $\mathcal{Y}$. On the other hand, if $x$ is random noise, there is heterogeneity but no meaningful connection between $\mathcal{X}$ and $\mathcal{Y}$, making non-trivial learning on $\mathcal{X}$ alone impractical.

The theory also highlights that and effective connections between modalities via generative models can enhance multimodal learning. This forms the basis for our Mixture of Noise Levels (MoNL) approach, which is particularly suited for multimodal learning with audio and video data.

## G.2 Advantages of Mixture of Noise Level Training

Our MoNL training method offers significant benefits for multimodal learning, especially with audio and video data:

**Heterogeneity and connection.** Audio and video are naturally heterogeneous. For example, a video of a person speaking includes audio of spoken words and video of lip movements and facial expressions. MoNL uses variable noise levels to enhance learning by capturing the generic transition matrix across the temporal axis.

$$p_\theta\big([\mathbf{z}_{t^{(1,1)}-1}^{(1,1)}, \ldots, \mathbf{z}_{t^{(M,N)}-1}^{(M,N)}] \mid [\mathbf{z}_{t^{(1,1)}}^{(1,1)}, \ldots, \mathbf{z}_{t^{(M,N)}}^{(M,N)}]\big) \qquad \text{(Eq. 4)}$$

**Enhanced connectivity.** MoNL improves connectivity between audio and video modalities. Our experiments show that MoNL often surpasses task-specific learning approaches by fostering better connections between modalities, adapting its focus more effectively.

## G.3 Enhanced Connectivity - Comparison with Existing Methods

**MoNL vs. Joint learning in MMD [40].** Unlike joint learning methods that focus on the joint distribution $p_\theta(\mathbf{z}_{t-1} \mid \mathbf{z}_t)$, MoNL trains across multiple conditioning, enabling better connections by varying its focus. This is evidenced by MoNL outperforming the Vanilla (see Table 1) and MMD models (see Tables 2 and 3).

**MoNL vs. Per-modality training.** MoNL goes beyond per-modality training in UniDiffuser [3], which uses variable noise between modalities i.e., learning $p_\theta\big([\mathbf{z}_{t^{(1)}-1}^{(1)}, \ldots, \mathbf{z}_{t^{(M)}-1}^{(M)}] \mid [\mathbf{z}_{t^{(1)}}^{(1)}, \ldots, \mathbf{z}_{t^{(M)}}^{(M)}]\big)$. MoNL introduces variable noise across different time segments, learning connections across temporal dynamics as well. This advantage is demonstrated in Table 1.

**MoNL vs. Masked training [49].** Diffusion models often obscure high-frequency details with low noise and low-frequency structures with high noise [7]. MoNL employs variable noise levels to explore diverse frequency components, enhancing the model's ability to correlate high and low-frequency elements. This is in contrast to masked self-supervised learning, which limits frequency-specific connections by masking entire elements.

In summary, the effectiveness of MoNL for multimodal diffusion models, particularly with audio and video data, stems from its strategic use of connection and heterogeneity. By applying variable noise levels, MoNL enhances connectivity between modalities and better adapts to diverse temporal and frequency components, leading to superior performance compared to existing multimodal learning methods.

