# OpenReview forum: "A Versatile Diffusion Transformer with Mixture of Noise Levels for Audiovisual Generation"
_NeurIPS.cc/2024/Conference — NeurIPS 2024 poster_

### Official Review · Reviewer_4xRp · 2024-06-28

**Soundness:** 2
**Presentation:** 4
**Contribution:** 2
**Rating:** 5
**Confidence:** 4

**Summary:**

This paper presents a new diffusion model that is capable of generating data following various conditional distributions in multimodal data. Rather than applying a uniform noise level to all data elements, the proposed model permits the use of differing noise levels across modalities and time dimensions. This allows the model to concurrently learn a variety of conditional distributions, including cross-modal conditioning, temporal interpolation, and temporal continuation. The experimental results with audio-video datasets validate the effectiveness of the proposed model.

**Strengths:**

- The idea of MoNL is simple and reasonable. It definitely enables the model to effectively learn various conditional distributions simultaneously.

- The proposed method performs well in the experiments. MoNL particularly contributes to boost the performance of the model in AV-inpaint and AV-continue tasks.

- The manuscript is well-written and easy to follow.

**Weaknesses:**

- The major concern is on the experiments: the majority of the empirical evaluations are conducted with the internal dataset. As there are already several publicly-available datasets (such as Landscape, AIST++, and VGGSound) that are commonly used in the literature, I strongly encourage the authors to do at least the ablation studies with such datasets to make them reproducible.

- The novelty in methoology is somewhat limited. The idea of using various noise levels in multimodal generative modeling has been presented in the prior work [3]. A particular challenge in this paper is to extend this idea also for modeling temporal dynamics, but similar strategies are commonly adopted in several video generation studies:
    - “Diffusion Models for Video Prediction and Infilling,” TMLR 2022.
    - “MCVD: Masked Conditional Video Diffusion for Prediction, Generation, and Interpolation,” NeurIPS 2022.

**Questions:**

- In the ablations at Table 1, why is Vanilla not used? Without the Vanilla strategy, the model may not encounter cases where all elements are almost entirely noise or close to clean ones during training, which could potentially degrade the performance of the model to some extent. Therefore, a more reasonable approach for ablation might be to gradually add other strategies to Vanilla.

---

<post-rebuttal>

I have updated my rating from 4 to 5, as my concern on the dataset is somewhat resolved through the discussion.

**Limitations:**

Limitations are properly discussed in the appendix.

---

> ### Author Rebuttal · Authors · 2024-08-07
>
> > **[W1]**: *"The major concern is on the experiments: the majority of the empirical evaluations are conducted with the internal dataset. As there are already several publicly-available datasets (such as Landscape, AIST++, and VGGSound) that are commonly used in the literature, I strongly encourage the authors to do at least the ablation studies with such datasets to make them reproducible. "*
>
> While we agree that evaluating on a diverse set of datasets is crucial, we believe that our approach effectively addresses the core challenge of audiovisual alignment.
> Indeed, existing audiovisual datasets often lack the diversity of cues necessary for a comprehensive assessment of audiovisual synchorny.
>
> Our focus has been on developing a robust audiovisual generative model with the great alignment capable of handling diverse and challenging practical real-world scenarios such as talking avatar. The Monologues dataset (16M samples), with its extensive variety of audiovisual cues such as  human appearances(e.g., a range of perceived age, perceived gender expression, head pose, lighting), verbal(e.g., intonation, prosody, emotion), and nonverbal cues(e.g., head nods, body gestures and expressions), provides a strong foundation for evaluating our model's performance under these conditions. We believe that the insights gained from our experiments on this dataset are valuable contributions to the field.
>
> Furthremore, while we included AIST++(1020 samples) and Landscape(8233 samples) datasets as they are used in our baseline, MM-Diffusion [R1] for the evaluation, we found that AIST++ and Landscape datasets are very small datasets and the results from ablation studies may not be generalizable given the risk of overfitting and memorizing.
>
> We are committed to advancing the research in audiovisual generation and plan to opensource our Monologue datasets as well as explore the use of publicly available datasets with rich audiovisual cues.
>
> [R1] Ruan et al. "MM-Diffusion: Learning multi-modal diffusion models for joint audio and video generation." CVPR 2023
>
> $ $
>  > **[W2]**: *"The novelty in methoology is somewhat limited. The idea of using various noise levels in multimodal generative modeling has been presented in the prior work [3]. A particular challenge in this paper is to extend this idea also for modeling temporal dynamics, but similar strategies are commonly adopted in several video generation studies"*
>
> While the concept of noise levels in generative modeling has been explored, MoNL offers significant advancements. Unlike UniDiffuser [R2], which struggles with multimodal sequence data, MoNL introduces a generalized formulation for handling diverse modalities with temporal dynamics. Our systematic evaluation demonstrates MoNL's superior performance and versatility across various tasks, including audio-video continuation and interpolation.
>
> Unlike previous video-focused methods such as “Diffusion Models for Video Prediction and Infilling” and "MCVD", MoNL excels in handling cross-modal and noisy conditions. Our approach avoids complex masked input mechanisms and hyperparameter tuning, resulting in a simpler and more efficient model. These key distinctions highlight MoNL's novelty and contribution to the field.
>
> We will incorporate a more detailed comparison to related work, including quantitative performance metrics, in the revised paper.
>
> [R2] Bao et al. "One transformer fits all distributions in multi-modal diffusion at scale." ICML 2024
>
> $ $
>  > **[Q1]**: *"In the ablations at Table 1, why is Vanilla not used? Without the Vanilla strategy, the model may not encounter cases where all elements are almost entirely noise or close to clean ones during training, which could potentially degrade the performance of the model to some extent. Therefore, a more reasonable approach for ablation might be to gradually add other strategies to Vanilla."*
>
>
>
> While we appreciate the reviewer's suggestion, our ablation design was carefully considered. Our goal was to systematically demonstrate the contributions of different components to our model's overall performance.
>
> - **Vanilla as a Baseline**: Our "joint generation" model, $\texttt{Vanilla}$, serves as a baseline for comparison. This model represents a unconditional training for joint generation without specialized handling of multimodal or cross-modal tasks.
> - **Task-Specific Strategies**: We introduced $\texttt{Pt}$, $\texttt{Pm}$, and $\texttt{Ptm}$ to address specific challenges in multimodal and cross-modal generation. Our ablation compares these strategies against the baseline to quantify their impact.
> - **Combined Approach**: The $\texttt{Pt/Pm/Ptm}$ model demonstrates the effectiveness of combining task-specific strategies.
> - **MoNL with $\texttt{Vanilla}$**: Finally, we compare MoNL (which incorporates $\texttt{Vanilla}$) to $\texttt{Pt/Pm/Ptm}$ to highlight the additional benefits of the $\texttt{Vanilla}$ components for joint generation.
>
> Furthermore, we found that even though the model may not encounter cases where all elements are almost entirely noise or close to clean ones during training, each strategy of $\texttt{Pt}$, $\texttt{Pm}$, and $\texttt{Ptm}$ can work overally and excels at each target task.

---

> > ### Comment · Reviewer_4xRp · 2024-08-08
> > **Thanks for the response**
> >
> > Thanks for the response. I have read the response as well as the other reviews.
> >
> > I have considered the author's rebuttal, but it does not provide new or compelling information that would change my evaluation. Therefore, I will maintain my current score.

---

> ### Author Response · Authors · 2024-08-12
> **Theoretical Background on Mixture of Noise Levels**
>
> ## Theoretical Background on Mixture of Noise Levels
> $ $
> ### 1. Theoretical Background on Multimodal Learning
>
> In *"A Theory of Multimodal Learning"* [1], multimodal learning is shown to offer a superior **generalization bound** compared to unimodal learning, with an improvement factor of **$O(\sqrt{n})$**, where **$n$** denotes the sample size. This benefit relies on **connection** and **heterogeneity** between modalities:
>
> - **Connection**: The bound depends on learned connections between (**$\mathcal{X}$**) and (**$\mathcal{Y}$**).
> - **Heterogeneity**: Describes how modalities, **$\mathcal{X}$** and **$\mathcal{Y}$**,  diverge and complement.
>
> If connection and heterogeneity are missing, ill-conditioned scenarios can arise. For instance, if **$x \equiv y$**, perfect connection suggests no need for learning about **$\mathcal{Y}$**. On the other hand, if **$x$** is random noise, there is heterogeneity but no meaningful connection between **$\mathcal{X}$** and **$\mathcal{Y}$**, making non-trivial learning on **$\mathcal{X}$** alone impractical.
>
> The theory also highlights that and learning effective connections between modalities via **generative models** can enhance multimodal learning. This forms the basis for our **Mixture of Noise Levels (MoNL)** approach, which is particularly suited for multimodal learning with **audio** and **video** data.
>
> [1] Zhou Lu. *"A Theory of Multimodal Learning."* NeurIPS 2023.
>
> $ $
> ### 2. Advantages of Mixture of Noise Level Training (MoNL)
>
> Our **Mixture of Noise Level (MoNL)** training method offers significant benefits for multimodal learning, especially with **audio** and **video** data:
>
> - **Heterogeneity and Connection**: Audio and video are naturally heterogeneous. For example, a video of a person speaking includes **audio** of spoken words and **video** of lip movements and facial expressions. MoNL uses **variable noise levels** to enhance learning by capturing the **generic transition matrix** across the **temporal axis**.
> $$p _{\mathbf{\theta}}([\mathbf{z} _{t ^{(1,1)}-1} ^{(1,1)}, \ldots, \mathbf{z} _{t ^{(M, N)}-1} ^{(M, N)}] \mid[\mathbf{z} _{t ^{(1,1)}} ^{(1,1)}, \ldots, \mathbf{z} _{t ^{(M, N)}} ^{(M, N)}]) \qquad \text{(Eq. (4))}$$
> where $M$, $N$ are the number of modalities and time-segments, respectively.
>
> - **Enhanced Connectivity**: MoNL improves **connectivity** between **audio** and **video** modalities. Our experiments show that MoNL often surpasses **task-specific learning** approaches by fostering better connections between modalities, adapting its focus more effectively.
>
> $ $
> ### 3. Enhanced Connectivity - Comparison with Existing Methods
>
> - **MoNL vs. Joint Learning in MMD** [2]: Unlike joint learning methods that focus on the joint distribution $p_{\mathbf{\theta}}(\mathbf{z} _{t-1} \mid\mathbf{z} _{t})$, MoNL trains across **multiple conditioning**, enabling better connections by varying its focus. This is evidenced by MoNL outperforming the Vanilla (see Table 1) and MMD models (see Tables 2 and 3).
>
> - **MoNL vs. Per-Modality Training**: MoNL goes beyond per-modality training in UniDiffuser [3], which uses variable noise between modalities i.e., learning $p_{\mathbf{\theta}}([\mathbf{z} _{t ^{(1)}-1} ^{(1)}, \ldots, \mathbf{z} _{t ^{(M)}-1} ^{(M)}] \mid[\mathbf{z} _{t ^{(1)}} ^{(1)}, \ldots, \mathbf{z} _{t ^{(M)}} ^{(M)}])$. MoNL introduces variable noise across different **time segments**, learning connections across **temporal dynamics** as well. This advantage is demonstrated in Table 1.
>
> - **MoNL vs. Masked Training** [4]: Diffusion models often obscure high-frequency details with low noise and low-frequency structures with high noise [5]. MoNL employs variable noise levels to explore diverse **frequency components**, enhancing the model's ability to correlate high and low-frequency elements. This is in contrast to masked self-supervised learning, which limits frequency-specific connections by masking entire elements.
>
> [2] Ruan et al. *"MM-Diffusion: Learning Multi-Modal Diffusion Models for Joint Audio and Video Generation."* CVPR 2023.
>
> [3] Bao et al. *"One Transformer Fits All Distributions in Multi-Modal Diffusion at Scale."* ICML 2023.
>
> [4] Voleti et al. *"MCVD: Masked Conditional Video Diffusion for Prediction, Generation, and Interpolation."* NeurIPS 2022.
>
> [5] Sander Dieleman. *"Noise Schedules Considered Harmful."* [Link](https://sander.ai/2024/06/14/noise-schedules).
>
> $ $
> ### Conclusion
>
> In summary, the effectiveness of **MoNL** for **multimodal diffusion models**, particularly with **audio** and **video** data, stems from its strategic use of **connection** and **heterogeneity**. By applying **variable noise levels**, MoNL enhances **connectivity** between modalities and better adapts to diverse **temporal** and **frequency components**, leading to superior performance compared to existing multimodal learning methods.

---

> ### Author Response · Authors · 2024-08-12
> **Clarification on Resource Constraints and Additional Theoretical Contributions**
>
> Dear Reviewer 4xRp,
>
> Thank you for your feedback and for considering our initial response.
>
>
> We acknowledge the importance of validating our model on publicly available datasets. Unfortunately, training on these datasets with the necessary computational resources (requiring approximately $35k per model) presented a significant challenge at the moment. However, we plan to proceed the experiments as the permit is approved and believe our work with the Monologues dataset, which contains 16M samples with a diverse range of audiovisual cues, provides valuable insights and demonstrates the robustness of our approach.
>
> In addition, we have newly provided the **"Theoretical Background on Mixture of Noise Levels,"** which we hope you will consider as part of our ongoing effort to strengthen our work and contribute to the field.
>
> We hope you understand our constraints and continue to see the value in the contributions we have made with the resources available. We remain committed to advancing research in **audiovisual generation** and plan to release the Monologues dataset to further support reproducibility in the future and encourage future work.
>
> Thank you for your consideration!

---

> > ### Comment · Reviewer_4xRp · 2024-08-13
> > **Thanks for further clarification**
> >
> > I appreciate that the authors plan to release the Monologues dataset. I would like to hear more on how the dataset was properly constructed to avoid any issues related to license or private information.
> >
> > One quick comment on the additional theoretical background. This theoretical perspective itself sounds really interesting. While I have no doubt on heterogeneity of audio and video, it seems that it is not trivial to show how MoNL contributes to boost connectivity in theory.

---

> ### Author Response · Authors · 2024-08-13
> **Clarifications and Additional Insights on the Monologues Dataset and MoNL**
>
> Dear Reviewer 4xRp,
>
>
> Thank you for your valuable feedback. We appreciate your interest in the Monologues dataset and your insights into the theoretical aspects of our work.
>
>
>
>
> **Dataset Construction**
> To address your concerns about dataset construction, we want to emphasize that the Monologues dataset was meticulously curated to adhere to ethical and legal standards. We have implemented robust measures to protect user privacy and avoid copyright infringement. The dataset consists exclusively of public links (and timestamps) to videos that are:
> * Including those that are **older than 90 days**
> * Free from **copyright claims**
> * Excluding **harmful or inappropriate material** (e.g., nudity, violence)
> * Verified to contain **motion** (excluding static images)
> * In the **English language** with a **single visible speaker**
> * Framed with **torso up**
>
> The dataset link list is automatically updated on a daily basis to ensure ongoing compliance with these criteria.
>
> The Monologues dataset is a valuable resource for the research community, providing a rich audiovisual benchmark for developing and evaluating multimodal models.
>
>
>
> **Theoretical Background**
> While formally proving the connectivity benefits of MoNL presents a significant challenge, our empirical results provide compelling evidence of its effectiveness. Table 1 and Figures 2 and 7 demonstrate MoNL's superior performance across diverse audiovisual tasks, generating more temproally-consistent audiovisuals compared to joint learning and per-modality method, suggesting enhanced intra- and inter-modality connectivity.
>
> We recognize the need for a deeper theoretical understanding and plan to explore this area in future research. As highlighted in *"A Theory of Multimodal Learning."*  (Zhou et al. NeurIPS 2023), theoretical foundations in multimodal learning remain relatively underdeveloped. Our work serves as a **crucial experimental foundation** for future theoretical investigations in this field. We believe our work represents a **substantial step forward in audiovisual and multimodal research**.
>
>
> **We believe these clarifications provide valuable insights into our work and would be grateful for your consideration of an improved score.**
>
> Sincerely,
>
> The Authors

---

> ### Author Response · Authors · 2024-08-14
> **Additional analysis of the objectives and effectiveness of our approach**
>
> To enhance the understanding of how MoNL contributes to building a unified model for diverse audiovisual tasks—including cross-modal inference and multimodal interpolation—  and improve connectivity between modalities, we provide a detailed analysis of the objectives and effectiveness of our approach, considering a simplified case with two modalities and two time-segments.
>
> $ $
> ### Objective Functions
>
> To determine the optimal objective function for learning a model $\theta$ that estimates:
>
> - **Cross-modal inferences:** $P(X|Y)$ and $P(Y|X)$
> - **Multimodal interpolations:** $P([\mathbf{x}_2, \mathbf{y}_2] \mid [\mathbf{x}_1, \mathbf{y}_1])$, $P([\mathbf{x}_1, \mathbf{y}_1] \mid [\mathbf{x}_2, \mathbf{y}_2])$, $P([\mathbf{x}_1, \mathbf{y}_2] \mid [\mathbf{x}_2, \mathbf{y}_1])$, and $P([\mathbf{x}_2, \mathbf{y}_1] \mid [\mathbf{x}_1, \mathbf{y}_2])$
>
> we evaluate the following objectives for heterogeneous multimodal data $X$ and $Y$, where $X = [\mathbf{x}_1, \mathbf{x}_2]$ and $Y = [\mathbf{y}_1, \mathbf{y}_2]$.
>
> 1. **Task-Specific Method**
>
>    $$
>    \min \mathbb{E}[P_\theta(X \mid Y)]
>    $$
>
>    This approach focuses solely on one direction of inference, potentially missing significant interactions between modalities.
>
> 2. **Joint Learning Method**
>
>    $$
>    \min \mathbb{E}[P_\theta(X, Y)]
>    $$
>
>    This method captures both modalities simultaneously but may fail to model temporal and multimodal interactions, which are crucial for conditional tasks.
>
> 3. **Per-Modality Method**
>
>    $$
>    \min \mathbb{E}\left[P_\theta(X, Y) + P_\theta(X \mid Y) + \alpha P_\theta(Y \mid X)\right]
>    $$
>
>    This method improves upon the task-specific approach by incorporating reverse inference between modalities. However, it may not effectively capture temporal interactions, which are important for interpolation tasks.
>
> 4. **Mixture of Noise Levels (MoNL)**
>
>    $$
>    \min \mathbb{E}\left[P_\theta(X, Y) + \beta_1 P_\theta(X \mid Y) + \beta_2 P_\theta(Y \mid X) + \beta_3 P([\mathbf{x}_2, \mathbf{y}_2] \mid [\mathbf{x}_1, \mathbf{y}_1]) + \beta_4 P([\mathbf{x}_1, \mathbf{y}_1] \mid [\mathbf{x}_2, \mathbf{y}_2]) + \beta_5 P([\mathbf{x}_1, \mathbf{y}_2] \mid [\mathbf{x}_2, \mathbf{y}_1]) + \beta_6 P([\mathbf{x}_2, \mathbf{y}_1] \mid [\mathbf{x}_1, \mathbf{y}_2])\right]
>    $$
>
>    MoNL integrates joint distributions, cross-modal inferences, and multimodal interpolations by learning a transition matrix that captures complex interactions between modalities.
>
> $ $
> ### Heterogeneous and Connected Multimodality
>
> For heterogeneous multimodal data (e.g., audio and video), where $X$ and $Y$ are interconnected but distinct, MoNL excels by capturing intricate interactions through:
>
> - **Direct Relationships:** Modeled by $P_\theta(X, Y)$
> - **Cross-Modal Inferences:** Modeled by $P_\theta(X \mid Y)$ and $P_\theta(Y \mid X)$
> - **Multimodal Interpolations:** Modeled by $P([\mathbf{x}_2, \mathbf{y}_2] \mid [\mathbf{x}_1, \mathbf{y}_1])$, $P([\mathbf{x}_1, \mathbf{y}_1] \mid [\mathbf{x}_2, \mathbf{y}_2])$ and similar terms.
>
> MoNL effectively models the connectivity and complex interactions between modalities, significantly enhancing performance in various audiovisual tasks.
>
>
> $ $
> We hope this analysis is helpful and will be incorporated into the final version.

---

> > ### Comment · Reviewer_4xRp · 2024-08-14
> > **Thanks for the response**
> >
> > Thanks for the response. I will update my rating after the discussion with the other reviwers.

---

### Official Review · Reviewer_SA7V · 2024-07-01

**Soundness:** 3
**Presentation:** 2
**Contribution:** 2
**Rating:** 5
**Confidence:** 4

**Summary:**

The paper tackles the audio-visual cross-modality generation problem and proposes a training approach to learn arbitrary conditional distributions in the audiovisual space. At the methodological level, the authors propose to apply variable diffusion timesteps across the temporal dimension. The experiments are conducted on Monologues, AIST++, and Landscape datasets.

**Strengths:**

The high-level motivation to learn arbitrary conditional distributions in the audiovisual space is interesting. The experiments show promising results and the authors conduct user studies as supplementary evaluations.

**Weaknesses:**

1. The writing and presentation of the paper can be improved. Figure 1 seems to have some issues with the first-row caption for the AIST dataset, which makes it difficult to read. The usage of math symbols is inconsistent, e.g., the $x$ should be $\bf{x}$ in Line 72. The term “multivariate data” in Line 63 is also not rigorous and confusing, in other words, do the authors imply that the static image data is uni-variante? The use of $\equiv$ and $=$ is also mixed in the paper.

2. One of the key claims and motivations, “Training separate models for each variation is expensive and impractical”, seems controversial and susceptible to me, while I understand training separate DMs for audio and visual data could be expensive, I don’t think that learning a single mixture is a better option than learning two separate data distributions, because learning a mixture increase the complexity of target data distribution, and thus should not be beneficial for performance especially if the domain gap between separate modality is large. As this is one of the key claims that motivates the methodological design, I was expecting more rigorous theoretical support in the paper, could the authors elaborate on that?

3. For the experiments, while the authors claim that “the sequential representations can be either latent spaces or raw data”, the actual implementation only conducts experiments on low dimensional features of pre-trained models, i.e., MAGVIT-v2 for video representations and SoundStream for audio representations. While this is not a serious flaw, it raises questions about the generalizability of the proposed method, and the fairness in comparison with other methods, as the pre-trained models may (and very likely) directly influence the final performance. And the baseline method MM-Diffusion seems not to operate on the same feature space according to Appendix C?

**Questions:**

In addition to the comments in the Weaknesses section.
I may have missed several details: how is the scheduler defined in the proposed method Eq. (5)? What is the time cost for training the proposed model? And how does this design yield a difference compared to training on single modality or separate DMs?

**Limitations:**

The limitations are discussed in the Appendix A.

---

> ### Author Rebuttal · Authors · 2024-08-07
>
> > **[W1]**: *"The writing and presentation of the paper can be improved. Figure 1 seems to have some issues with the first-row caption for the AIST dataset, which makes it difficult to read. The usage of math symbols is inconsistent, e.g., the $𝑥$ should be  $\boldsymbol{x}$ in Line 72. The term “multivariate data” in Line 63 is also not rigorous and confusing, in other words, do the authors imply that the static image data is uni-variante? The use of ≡ and =  is also mixed in the paper."*
>
>
> We sincerely appreciate the reviewer’s constructive feedback on the clarity and presentation of our paper. We agree that improving these aspects is crucial for effective communication of our research.
>
> We will meticulously address the reviewer’s comments by enhancing the readability of Figure 1, ensuring consistency in mathematical notation (e.g., using $\boldsymbol{x}$ for vectors), and refining ambiguous terminology such as "multivariate data" to accurately reflect the sequential nature of our data. We will also carefully distinguish between the use of ≡ and = throughout the paper.
>
>
> $ $
>  > **[W2&Q3]**: *"One of the key claims and motivations, “Training separate models for each variation is expensive and impractical”, seems controversial and susceptible to me, while I understand training separate DMs for audio and visual data could be expensive, I don’t think that learning a single mixture is a better option than learning two separate data distributions, because learning a mixture increase the complexity of target data distribution, and thus should not be beneficial for performance especially if the domain gap between separate modality is large. As this is one of the key claims that motivates the methodological design, I was expecting more rigorous theoretical support in the paper, could the authors elaborate on that?"*
>
>
> We appreciate the reviewer's insightful comments. While learning a single mixture model can increase complexity compared to separate models, we argue that the benefits of a unified model outweigh the potential drawbacks, especially in the context of large-scale multimodal tasks.
>
> The reviewer points out the potential challenges of learning a mixture distribution with large domain gaps. While we acknowledge this, we believe that the shared underlying structure of audio and video data, such as temporal synchronization and physical world correlations, can help mitigate these issues. Our model leverages these shared representations to improve performance across various tasks. This is supported by our experimental results in Table 1 as well as the recent works on audio-visual representation learning (AVMAE [R1]) which demonstrates the benefits of multimodal learning on the audiovisual tasks compared to unimodal learning for audio and video respectively. Also, UniDiffuser [R2] show the success of multimodal generation in the text-image domain.
>
> As you mentioned, training separate models for each task variation can be prohibitively expensive, especially when considering the vast number of potential combinations in the audio-video domain. Our approach offers a more efficient and scalable solution by learning a single model capable of handling multiple tasks simultaneously.
>
> We believe that our model offers a promising approach to tackling the challenges of large-scale multimodal tasks.
>
> [R1] Bao et al. "One transformer fits all distributions in multi-modal diffusion at scale." ICML 2024
>
> [R2] Georgescu et al. "Audiovisual masked autoencoders." CVPR 2023
>
> $ $
>  > **[W3]**: *"For the experiments, while the authors claim that “the sequential representations can be either latent spaces or raw data”, the actual implementation only conducts experiments on low dimensional features of pre-trained models, i.e., MAGVIT-v2 for video representations and SoundStream for audio representations. While this is not a serious flaw, it raises questions about the generalizability of the proposed method, and the fairness in comparison with other methods, as the pre-trained models may (and very likely) directly influence the final performance. And the baseline method MM-Diffusion seems not to operate on the same feature space according to Appendix C? "*
>
> We appreciate the reviewer's insightful comments. Regarding generalizability, we acknowledge the limitation of our current experiments to latent spaces. While we believe our method is conceptually applicable to raw data, practical constraints such as computational resources from conducting extensive experiments in this setting. We will clarify this point in the modified version. Note that our training data was not seen during training of the audio and video autoencoders.
>
> MMD has their own autoencoders. To address the fairness of comparison, comparison of MMD with our proposed MoNL and MMD with original joint learning would be ideal. We encountered challenges in effectively integrating our diffusion timestep vector with the sparse multimodal attention module (RS-MMA) – a core component of the MMD architecture.
> To provide a more equitable comparison, we opted to train a transformer-based variant of MMD (Vanilla in Table 1) as a counterpart to our transformer-based MoNL. Our results demonstrate the superiority of MoNL over this transformer-based MMD, highlighting the effectiveness of our proposed approach. Additionally, we directly compare AVDiT, MoNL, and MMD to further underscore the overall strength of our model.
>
> $ $
>  > **[Q1]**: *"how is the scheduler defined in the proposed method Eq. (5)? "*
>
> The noise scheduler used in Equation (5) is identical to the one detailed in Section 2 (Lines 68-74).
>
> $ $
>  > **[Q2]**: *"What is the time cost for training the proposed model?"*
>
> A comprehensive analysis of the training time costs is provided in the Supplementary Material, Section B (Lines 536-541). On average, the models were trained for around 350K steps with a batch size of 256 for around five days.

---

> > ### Comment · Reviewer_SA7V · 2024-08-09
> > **Post-rebuttal**
> >
> > I appreciate the author's efforts in preparing the rebuttal.
> > After reading the rebuttal, I think some of my concerns related to the experiments are clarified.
> > On the other hand, this is a rather empirical paper, and I cannot always find the underlying theoretical intuition of this entire line of work. However, maybe the brute-force scaling up can overpass the underlying theoretical justification over the challenge of learning disjoint distributions without much understanding of the data distribution and it seems to work fine.
> > Anyway, I raise my score to 5 for post-rebuttal.

---

> ### Author Response · Authors · 2024-08-12
> **Theoretical Background on Mixture of Noise Levels**
>
> ## Theoretical Background on Mixture of Noise Levels
> $ $
> ### 1. Theoretical Background on Multimodal Learning
>
> In *"A Theory of Multimodal Learning"* [1], multimodal learning is shown to offer a superior **generalization bound** compared to unimodal learning, with an improvement factor of **$O(\sqrt{n})$**, where **$n$** denotes the sample size. This benefit relies on **connection** and **heterogeneity** between modalities:
>
> - **Connection**: The bound depends on learned connections between (**$\mathcal{X}$**) and (**$\mathcal{Y}$**).
> - **Heterogeneity**: Describes how modalities, **$\mathcal{X}$** and **$\mathcal{Y}$**,  diverge and complement.
>
> If connection and heterogeneity are missing, ill-conditioned scenarios can arise. For instance, if **$x \equiv y$**, perfect connection suggests no need for learning about **$\mathcal{Y}$**. On the other hand, if **$x$** is random noise, there is heterogeneity but no meaningful connection between **$\mathcal{X}$** and **$\mathcal{Y}$**, making non-trivial learning on **$\mathcal{X}$** alone impractical.
>
> The theory also highlights that and learning effective connections between modalities via **generative models** can enhance multimodal learning. This forms the basis for our **Mixture of Noise Levels (MoNL)** approach, which is particularly suited for multimodal learning with **audio** and **video** data.
>
> [1] Zhou Lu. *"A Theory of Multimodal Learning."* NeurIPS 2023.
>
> $ $
> ### 2. Advantages of Mixture of Noise Level Training (MoNL)
>
> Our **Mixture of Noise Level (MoNL)** training method offers significant benefits for multimodal learning, especially with **audio** and **video** data:
>
> - **Heterogeneity and Connection**: Audio and video are naturally heterogeneous. For example, a video of a person speaking includes **audio** of spoken words and **video** of lip movements and facial expressions. MoNL uses **variable noise levels** to enhance learning by capturing the **generic transition matrix** across the **temporal axis**.
> $$p _{\mathbf{\theta}}([\mathbf{z} _{t ^{(1,1)}-1} ^{(1,1)}, \ldots, \mathbf{z} _{t ^{(M, N)}-1} ^{(M, N)}] \mid[\mathbf{z} _{t ^{(1,1)}} ^{(1,1)}, \ldots, \mathbf{z} _{t ^{(M, N)}} ^{(M, N)}]) \qquad \text{(Eq. (4))}$$
> where $M$, $N$ are the number of modalities and time-segments, respectively.
>
> - **Enhanced Connectivity**: MoNL improves **connectivity** between **audio** and **video** modalities. Our experiments show that MoNL often surpasses **task-specific learning** approaches by fostering better connections between modalities, adapting its focus more effectively.
>
> $ $
> ### 3. Enhanced Connectivity - Comparison with Existing Methods
>
> - **MoNL vs. Joint Learning in MMD** [2]: Unlike joint learning methods that focus on the joint distribution $p_{\mathbf{\theta}}(\mathbf{z} _{t-1} \mid\mathbf{z} _{t})$, MoNL trains across **multiple conditioning**, enabling better connections by varying its focus. This is evidenced by MoNL outperforming the Vanilla (see Table 1) and MMD models (see Tables 2 and 3).
>
> - **MoNL vs. Per-Modality Training**: MoNL goes beyond per-modality training in UniDiffuser [3], which uses variable noise between modalities i.e., learning $p_{\mathbf{\theta}}([\mathbf{z} _{t ^{(1)}-1} ^{(1)}, \ldots, \mathbf{z} _{t ^{(M)}-1} ^{(M)}] \mid[\mathbf{z} _{t ^{(1)}} ^{(1)}, \ldots, \mathbf{z} _{t ^{(M)}} ^{(M)}])$. MoNL introduces variable noise across different **time segments**, learning connections across **temporal dynamics** as well. This advantage is demonstrated in Table 1.
>
> - **MoNL vs. Masked Training** [4]: Diffusion models often obscure high-frequency details with low noise and low-frequency structures with high noise [5]. MoNL employs variable noise levels to explore diverse **frequency components**, enhancing the model's ability to correlate high and low-frequency elements. This is in contrast to masked self-supervised learning, which limits frequency-specific connections by masking entire elements.
>
> [2] Ruan et al. *"MM-Diffusion: Learning Multi-Modal Diffusion Models for Joint Audio and Video Generation."* CVPR 2023.
>
> [3] Bao et al. *"One Transformer Fits All Distributions in Multi-Modal Diffusion at Scale."* ICML 2023.
>
> [4] Voleti et al. *"MCVD: Masked Conditional Video Diffusion for Prediction, Generation, and Interpolation."* NeurIPS 2022.
>
> [5] Sander Dieleman. *"Noise Schedules Considered Harmful."* [Link](https://sander.ai/2024/06/14/noise-schedules).
>
> $ $
> ### Conclusion
>
> In summary, the effectiveness of **MoNL** for **multimodal diffusion models**, particularly with **audio** and **video** data, stems from its strategic use of **connection** and **heterogeneity**. By applying **variable noise levels**, MoNL enhances **connectivity** between modalities and better adapts to diverse **temporal** and **frequency components**, leading to superior performance compared to existing multimodal learning methods.

---

> > ### Author Response · Authors · 2024-08-12
> > **Follow-Up on Theoretical Clarifications and Request for Final Review**
> >
> > Dear Reviewer SA7V,
> >
> > Thank you for your feedback and for raising your score after reviewing our rebuttal. We appreciate your acknowledgment of our efforts to clarify the experimental aspects of our work.
> >
> > In response to your concern about the theoretical intuition, we want to highlight that we have newly provided the **"Theoretical Background on Mixture of Noise Levels,"** which we hope you will consider as part of our ongoing effort to strengthen our work and contribute to the field.
> >
> > As we have only two days left for further discussion, we kindly request that you review our additional responses. We sincerely appreciate the time and effort you have dedicated to reviewing our paper and your constructive and insightful comments.
> >
> > Thank you once again.
> >
> > Best regards,
> > The Authors

---

### Official Review · Reviewer_tkoE · 2024-07-12

**Soundness:** 3
**Presentation:** 2
**Contribution:** 3
**Rating:** 5
**Confidence:** 4

**Summary:**

This paper introduces the Audiovisual Diffusion Transformer (AVDiT) with Mixture of Noise Levels (MoNL) for audiovisual sequence generation. The key innovation is the use of variable noise levels during the diffusion process, applied across different time segments and modalities. This approach enables the model to effectively learn arbitrary conditional distributions in a task-agnostic manner, making it versatile for various generation tasks such as cross-modal generation, multimodal interpolation, and audiovisual continuation. Experiments on multiple datasets demonstrate that AVDiT with MoNL outperforms existing baselines, generating temporally and perceptually consistent audiovisual sequences.

**Strengths:**

1. The proposed variable noise levels across different time segments and modalities is an effective approach to enhance the flexibility of diffusion models.
2. The model's ability to handle various audiovisual generation tasks within a single framework is impressive.
3. The paper includes extensive experiments on multiple datasets, with both qualitative and quantitative evaluations. The additional demo page provides an intuitive comparison.

**Weaknesses:**

1. While the variable noise levels concept is interesting, the overall novelty of the approach may be seen as incremental. Similar techniques in diffusion models and transformers have been explored in papers like MM-Diffusion.
2. Some technical details are not thoroughly explained, such as the criteria for selecting and varying noise levels across time segments and modalities.
3. The evaluation is primarily conducted on a few datasets (Monologues, AIST++, Landscape). The model's generalizability to real-world scenarios remains uncertain.
4. The authors compare the transformer-based AVDiT model to the UNet-based MM-Diffusion model, which might not be a fair comparison due to the different architectures. The authors should consider training a UNet-based model using their proposed approach to provide a more direct comparison and validate the effectiveness of their method.

**Questions:**

See weakness.

**Limitations:**

The authors mention some limitations in the appendix, such as the need for further improvements in visual and speech quality and the potential for ethical concerns. However, for me, the lack of extensive evaluation of more diverse and real-world datasets is a noteworthy limitation. Some additional limitations can be found in the Weaknesses section.

---

> ### Author Rebuttal · Authors · 2024-08-07
>
> > **[W1]**: *"While the variable noise levels concept is interesting, the overall novelty of the approach may be seen as incremental. "*
>
> While we appreciate the reviewer's acknowledgment of the variable noise levels concept, we believe that our work offers significant advancements beyond prior art.
>
> - **Temporal Dynamics**: Unlike MM-Diffusion [R1], which struggles with generating temporally consistent sequences, our method effectively models temporal dependencies through AVDiT with MoNL. This enables superior performance on tasks requiring temporal coherence, such as multimodal interpolation and audiovisual continuation.
> - **Generalization and Flexibility**: Our approach is designed to handle a wide range of multimodal, sequential tasks, surpassing the limitations of previous methods like UniDiffuser [R2] and Versatile Diffusion [R3], which primarily focus on unimodal or static data. By introducing a generalized framework, we enable the modeling of complex audiovisual interactions and the generation of expressive, controllable multimedia content.
> - **Systematic Evaluation and Performance**: Our extensive evaluation demonstrates that AVDiT with MoNL consistently outperforms state-of-the-art baselines in generating high-quality, temporally coherent audiovisual sequences.
> - **Pioneering Diffusion Transformer for Audio-Video**: To the best of our knowledge, our work is the first to successfully apply a diffusion transformer to the audio-video multimodal domain. This novel architecture, combined with MoNL, has led to significant advancements in audiovisual sequence generation.
>
> [R1] Ruan et al. "MM-Diffusion: Learning multi-modal diffusion models for joint audio and video generation." CVPR 2023
>
> [R2] Bao et al. "One transformer fits all distributions in multi-modal diffusion at scale." ICML 2024
>
> [R3] Xuet al. "Versatile diffusion: Text, images and variations all in one diffusion model." CVPR 2023
>
> $ $
>  > **[W2]**: *"Some technical details are not thoroughly explained, such as the criteria for selecting and varying noise levels across time segments and modalities. "*
>
> We appreciate the reviewer's keen interest in the technical details of our work. Section 3.2 and Algorithm 1 provides a detailed explanation of how we selected and varied noise levels across time segments and modalities. For instance,  in MoNL that is a training paradigm where a timestep is uniformly randomly selected from the mixture, we first chosse a stratgy for selecting variable noise levels from $\mathcal{U}( \set{ \texttt {Vanilla}, \texttt {Pt}, \texttt {Pm}, \texttt {Ptm} } )$,  and the noise level is set by setting time step vectors depending on the type of the stratge following Line 123-135.
>
> We understand the importance of these technical details for reproducibility and have included comprehensive supplementary materials detailing implementation details on autoencoders, AVDiT, diffusion training, inference. Also we provided experimental setup and evaluation details. To enhance readability, we are committed to carefully selecting key details for inclusion in the main paper, ensuring that the core methodology is clear and accessible to a broad audience.
>
>
> $ $
>  > **[W3]**: *"The evaluation is primarily conducted on a few datasets (Monologues, AIST++, Landscape). The model's generalizability to real-world scenarios remains uncertain."*
>
> While we agree that evaluating on a diverse set of datasets is crucial, we believe that our approach effectively addresses the core challenge of audiovisual alignment.
> Indeed, existing audiovisual datasets often lack the diversity of cues necessary for a comprehensive assessment of audiovisual synchorny.
> The Monologues dataset (16M), with its rich variety of audiovisual cues provides a strong foundation for assessing this alignment: human appearances(e.g., a range of perceived age, perceived gender expression, head pose, lighting), verbal(e.g., intonation, prosody, emotion), and nonverbal cues(e.g., head nods, body gestures and expressions).
>
> To further demonstrate the generalizability of our model, we conducted comprehensive experiments on multiple datasets, including AIST++ and Landscape where our baseline evaluated. These datasets represent diverse audiovisual domains, ensuring a wide range of audio (music, speech, natural sounds) and video content (monologues, dance, natural scenes).
> This diverse evaluation demonstrates the versatility of our model in handling various audiovisual combinations.
>
>
> $ $
>  > **[W4]**: *"The authors compare the transformer-based AVDiT model to the UNet-based MM-Diffusion model, which might not be a fair comparison due to the different architectures. The authors should consider training a UNet-based model using their proposed approach to provide a more direct comparison and validate the effectiveness of their method. "*
>
>
> We appreciate the reviewer's insightful comment regarding the architectural differences between our transformer-based MoNL and the UNet-based MMD. While a comparison of MMD with our proposed MoNL and MMD with original joint learning would be ideal, we encountered challenges in effectively integrating our diffusion timestep vector with the sparse multimodal attention module (RS-MMA) – a core component of the MMD architecture.
>
> To provide a more equitable comparison, we opted to train a transformer-based joint learning that can be regarded as a variant of MMD (Vanilla in Table 1) as a counterpart to our transformer-based MoNL. Our results demonstrate the superiority of MoNL over this transformer-based MMD, highlighting the effectiveness of our proposed approach. Additionally, we directly compare AVDiT, MoNL, and MMD to further underscore the overall strength of our model.
>
> We will clarify this point and provide additional details in the revised manuscript.

---

> ### Author Response · Authors · 2024-08-12
> **Theoretical Background on Mixture of Noise Levels**
>
> ## Theoretical Background on Mixture of Noise Levels
> $ $
> ### 1. Theoretical Background on Multimodal Learning
>
> In *"A Theory of Multimodal Learning"* [1], multimodal learning is shown to offer a superior **generalization bound** compared to unimodal learning, with an improvement factor of **$O(\sqrt{n})$**, where **$n$** denotes the sample size. This benefit relies on **connection** and **heterogeneity** between modalities:
>
> - **Connection**: The bound depends on learned connections between (**$\mathcal{X}$**) and (**$\mathcal{Y}$**).
> - **Heterogeneity**: Describes how modalities, **$\mathcal{X}$** and **$\mathcal{Y}$**,  diverge and complement.
>
> If connection and heterogeneity are missing, ill-conditioned scenarios can arise. For instance, if **$x \equiv y$**, perfect connection suggests no need for learning about **$\mathcal{Y}$**. On the other hand, if **$x$** is random noise, there is heterogeneity but no meaningful connection between **$\mathcal{X}$** and **$\mathcal{Y}$**, making non-trivial learning on **$\mathcal{X}$** alone impractical.
>
> The theory also highlights that and learning effective connections between modalities via **generative models** can enhance multimodal learning. This forms the basis for our **Mixture of Noise Levels (MoNL)** approach, which is particularly suited for multimodal learning with **audio** and **video** data.
>
> [1] Zhou Lu. *"A Theory of Multimodal Learning."* NeurIPS 2023.
>
> $ $
> ### 2. Advantages of Mixture of Noise Level Training (MoNL)
>
> Our **Mixture of Noise Level (MoNL)** training method offers significant benefits for multimodal learning, especially with **audio** and **video** data:
>
> - **Heterogeneity and Connection**: Audio and video are naturally heterogeneous. For example, a video of a person speaking includes **audio** of spoken words and **video** of lip movements and facial expressions. MoNL uses **variable noise levels** to enhance learning by capturing the **generic transition matrix** across the **temporal axis**.
> $$p _{\mathbf{\theta}}([\mathbf{z} _{t ^{(1,1)}-1} ^{(1,1)}, \ldots, \mathbf{z} _{t ^{(M, N)}-1} ^{(M, N)}] \mid[\mathbf{z} _{t ^{(1,1)}} ^{(1,1)}, \ldots, \mathbf{z} _{t ^{(M, N)}} ^{(M, N)}]) \qquad \text{(Eq. (4))}$$
> where $M$, $N$ are the number of modalities and time-segments, respectively.
>
> - **Enhanced Connectivity**: MoNL improves **connectivity** between **audio** and **video** modalities. Our experiments show that MoNL often surpasses **task-specific learning** approaches by fostering better connections between modalities, adapting its focus more effectively.
>
> $ $
> ### 3. Enhanced Connectivity - Comparison with Existing Methods
>
> - **MoNL vs. Joint Learning in MMD** [2]: Unlike joint learning methods that focus on the joint distribution $p_{\mathbf{\theta}}(\mathbf{z} _{t-1} \mid\mathbf{z} _{t})$, MoNL trains across **multiple conditioning**, enabling better connections by varying its focus. This is evidenced by MoNL outperforming the Vanilla (see Table 1) and MMD models (see Tables 2 and 3).
>
> - **MoNL vs. Per-Modality Training**: MoNL goes beyond per-modality training in UniDiffuser [3], which uses variable noise between modalities i.e., learning $p_{\mathbf{\theta}}([\mathbf{z} _{t ^{(1)}-1} ^{(1)}, \ldots, \mathbf{z} _{t ^{(M)}-1} ^{(M)}] \mid[\mathbf{z} _{t ^{(1)}} ^{(1)}, \ldots, \mathbf{z} _{t ^{(M)}} ^{(M)}])$. MoNL introduces variable noise across different **time segments**, learning connections across **temporal dynamics** as well. This advantage is demonstrated in Table 1.
>
> - **MoNL vs. Masked Training** [4]: Diffusion models often obscure high-frequency details with low noise and low-frequency structures with high noise [5]. MoNL employs variable noise levels to explore diverse **frequency components**, enhancing the model's ability to correlate high and low-frequency elements. This is in contrast to masked self-supervised learning, which limits frequency-specific connections by masking entire elements.
>
> [2] Ruan et al. *"MM-Diffusion: Learning Multi-Modal Diffusion Models for Joint Audio and Video Generation."* CVPR 2023.
>
> [3] Bao et al. *"One Transformer Fits All Distributions in Multi-Modal Diffusion at Scale."* ICML 2023.
>
> [4] Voleti et al. *"MCVD: Masked Conditional Video Diffusion for Prediction, Generation, and Interpolation."* NeurIPS 2022.
>
> [5] Sander Dieleman. *"Noise Schedules Considered Harmful."* [Link](https://sander.ai/2024/06/14/noise-schedules).
>
> $ $
> ### Conclusion
>
> In summary, the effectiveness of **MoNL** for **multimodal diffusion models**, particularly with **audio** and **video** data, stems from its strategic use of **connection** and **heterogeneity**. By applying **variable noise levels**, MoNL enhances **connectivity** between modalities and better adapts to diverse **temporal** and **frequency components**, leading to superior performance compared to existing multimodal learning methods.

---

> ### Author Response · Authors · 2024-08-12
> **Request for Final Review: Response and Theoretical Clarifications**
>
> Dear Reviewer tkoE,
>
>
> We kindly request that you review our responses, as we have only **two days** left for further **discussion**. We have addressed your comments thoughtfully and provided a thorough **theoretical background** of our method. We sincerely appreciate the time and effort you have dedicated to reviewing our paper and your constructive and insightful feedback.
>
> Thank you once again.
>
> Best regards,
> The Authors

---

> > ### Comment · Reviewer_tkoE · 2024-08-12
> >
> > Thank you for addressing all my comments. Overall, it is a good paper on audiovisual generation, although I find the proposed method to be somewhat incremental. I suggest the authors make their trained models and code available for reproducibility. I'm willing to increase my rating to BA.

---

> ### Author Response · Authors · 2024-08-13
> **Thank you for your time and effort**
>
> Dear Reviewer tkoE,
>
> We sincerely appreciate your time and effort in reviewing our paper. We are glad that you found our work to be a good contribution to the field of audiovisual generation.
>
> We believe that our AVDiT with MoNL effectively models temporal dependencies, handles diverse multimodal tasks, and demonstrates superior performance compared to existing approaches, offering significant advancements beyond prior art. We believe these contributions can be valuable to the community!
>
> We are committed to fostering reproducibility in our research and are happy to make our trained models and code publicly available upon acceptance of the paper.
>
> Thank you again for your valuable insights.
>
> Sincerely,
>
> The Authors

---

### Official Review · Reviewer_bNvz · 2024-07-29

**Soundness:** 2
**Presentation:** 3
**Contribution:** 2
**Rating:** 6
**Confidence:** 3

**Summary:**

The paper presents a new method for audiovisual generation where the input output condiitions may comprise of 2 modalities, namely video and audio sequence. The authors propose a new training approach to effectively learn conditional distribution in multimodal space. The main novely in the paper is a mixture of noise level formulation for processing the inputs. The method produces temporally consistent samples and outperforms existing arts and the vanilla configuration

**Strengths:**

1. The utilization of mixture of noise levels is novel and the methods seems to improve robustness in denoising, hence leading to a performance boost
2. The evaluations seems to be fair and clearly signifies the working of the method.
3. The paper is well written and easy to follow.

**Weaknesses:**

1. Although the method seems to work well empirically, the paper lacks theoretical backing for the proposed method. It would be good to see some proofs that the proposed method leads to a better approximation of the variational lower bound and joint distribution.
2. Are there any sampling modifications required to accommodate the proposed training strategy?
3. There is a plethora of compositional works derived from the energy based formulation of diffusion models. Could the authors analyze how the proposed method performs in comparison to it.
[1] https://energy-based-model.github.io/Compositional-Visual-Generation-with-Composable-Diffusion-Models/
4. I'm rating the paper borderline for now, I will improve my rating if the authors can address my concerns

**Questions:**

1. Could the authors provide a proof of why the proposed method would work better
2. Are there any sampling modifications required to accommodate the proposed training strategy?
3. Could the authors give an analysis of how the proposed algorithm will work when compared to [1]

**Limitations:**

Please see weakness.

---

> ### Author Rebuttal · Authors · 2024-08-07
>
> > **[W1&Q1]**: *"Although the method seems to work well empirically, the paper lacks theoretical backing for the proposed method. It would be good to see some proofs that the proposed method leads to a better approximation of the variational lower bound and joint distribution."*
>
> We appreciate the reviewer's request for theoretical justification. Our approach leverages variable noise levels to learn a general transition matrix (Eq. 4, Section 3.1) between modalities and time segments. This can be used for the inference of arbitrary conditional distributions with various input-output combinations and enables handling diverse conditional tasks (Section 3.3).
>
>
> To elucidate the core idea, let's consider a simplified scenario where the multivariate data consists of only two elements, $x$ and $y$. In this case, each transition matrix can be interpreted as learning a conditional distribution of the form $p(x|y=c)$, where $c$ represents a specific condition. By training across multiple variable noise levels, our objective become as follows:
>
> $$L = E_{x,y}[-\text{log} \ p(x,y)] + α E_{x,y}[-\text{log} \ p(x|y)]$$
>
> where $α$ balances the importance of joint and conditional learning depending on the variable noise schedule stratgies. This formulation demonstrates that our method simultaneously learns the joint distribution and conditional distributions, offering potential advantages over methods that solely focus on joint learning $E_{x,y}[-\text{log} \ p(x,y)]$, such as MM-Diffusion.
>
> While this simplified explanation provides intuition, we acknowledge that this is a simplified explanation and that a more rigorous theoretical analysis is ongoing. We plan to provide a comprehensive proof and generalized formulation in future work.
>
> $ $
>  > **[W2&Q2]**: *"Are there any sampling modifications required to accommodate the proposed training strategy? "*
>
> We detailed sampling modifications in Section 3.3 and Figure 4. Inference involves selective noise injection based on the task. Clean inputs are used for conditioned parts ($t^{(m, n)} = 0$), while noisy inputs are injected for desired generation ($t^{(m, n)} = t$). This allows a single model to handle various conditional tasks with diverse input-output combinations.
>
>
> $ $
>  > **[W3&Q3]**: *"There is a plethora of compositional works derived from the energy based formulation of diffusion models. Could the authors analyze how the proposed method performs in comparison to it. [1] "*
>
> Both methods involve energy-based diffusion models. However, our work differs significantly:
> - **Target Tasks/Modality**: We focus on Audio-to-Video, Video-to-Audio, and Audiovisual continuation and interpolation tasks with variable durations. Composable Diffusion targets only Text-to-Image generation.
> - **Objective of Method**: Composable Diffusion focuses on composing concepts at inference without further training ($p(\boldsymbol{x}|\boldsymbol{y}^{(1)}, ..., \boldsymbol{y}^{(C)})$ where $c_i \in [1,C]$ represents each concept). We aim for a single model handling diverse conditional distributions ($p(\boldsymbol{x}_0^{(n \in \mathcal{N}_x^c)}, \boldsymbol{y}_0^{(n \in \mathcal{N}_y^c)} \mid \boldsymbol{x}_0^{(n \in \mathcal{N}_x)}, \boldsymbol{y}_0^{(n \in \mathcal{N}_y)})$ where
>  $\mathcal{N}_x$ and $\mathcal{N}_y$ are input index sets).
> - **Training/Inference**: Composable Diffusion uses weighted summation of conditional scores during inference. We learn a general transition matrix during training and perform selective noise injection based on the task for inference.
>
> These fundamental differences highlight that our methods can be complementary. We envision combining them for complex tasks like predicting background audio from two videos.

---

> ### Author Response · Authors · 2024-08-12
> **Theoretical Background on Mixture of Noise Levels**
>
> ## Theoretical Background on Mixture of Noise Levels
> $ $
> ### 1. Theoretical Background on Multimodal Learning
>
> In *"A Theory of Multimodal Learning"* [1], multimodal learning is shown to offer a superior **generalization bound** compared to unimodal learning, with an improvement factor of **$O(\sqrt{n})$**, where **$n$** denotes the sample size. This benefit relies on **connection** and **heterogeneity** between modalities:
>
> - **Connection**: The bound depends on learned connections between (**$\mathcal{X}$**) and (**$\mathcal{Y}$**).
> - **Heterogeneity**: Describes how modalities, **$\mathcal{X}$** and **$\mathcal{Y}$**,  diverge and complement.
>
> If connection and heterogeneity are missing, ill-conditioned scenarios can arise. For instance, if **$x \equiv y$**, perfect connection suggests no need for learning about **$\mathcal{Y}$**. On the other hand, if **$x$** is random noise, there is heterogeneity but no meaningful connection between **$\mathcal{X}$** and **$\mathcal{Y}$**, making non-trivial learning on **$\mathcal{X}$** alone impractical.
>
> The theory also highlights that and learning effective connections between modalities via **generative models** can enhance multimodal learning. This forms the basis for our **Mixture of Noise Levels (MoNL)** approach, which is particularly suited for multimodal learning with **audio** and **video** data.
>
> [1] Zhou Lu. *"A Theory of Multimodal Learning."* NeurIPS 2023.
>
> $ $
> ### 2. Advantages of Mixture of Noise Level Training (MoNL)
>
> Our **Mixture of Noise Level (MoNL)** training method offers significant benefits for multimodal learning, especially with **audio** and **video** data:
>
> - **Heterogeneity and Connection**: Audio and video are naturally heterogeneous. For example, a video of a person speaking includes **audio** of spoken words and **video** of lip movements and facial expressions. MoNL uses **variable noise levels** to enhance learning by capturing the **generic transition matrix** across the **temporal axis**.
> $$p _{\mathbf{\theta}}([\mathbf{z} _{t ^{(1,1)}-1} ^{(1,1)}, \ldots, \mathbf{z} _{t ^{(M, N)}-1} ^{(M, N)}] \mid[\mathbf{z} _{t ^{(1,1)}} ^{(1,1)}, \ldots, \mathbf{z} _{t ^{(M, N)}} ^{(M, N)}]) \qquad \text{(Eq. (4))}$$
> where $M$, $N$ are the number of modalities and time-segments, respectively.
>
> - **Enhanced Connectivity**: MoNL improves **connectivity** between **audio** and **video** modalities. Our experiments show that MoNL often surpasses **task-specific learning** approaches by fostering better connections between modalities, adapting its focus more effectively.
>
> $ $
> ### 3. Enhanced Connectivity - Comparison with Existing Methods
>
> - **MoNL vs. Joint Learning in MMD** [2]: Unlike joint learning methods that focus on the joint distribution $p_{\mathbf{\theta}}(\mathbf{z} _{t-1} \mid\mathbf{z} _{t})$, MoNL trains across **multiple conditioning**, enabling better connections by varying its focus. This is evidenced by MoNL outperforming the Vanilla (see Table 1) and MMD models (see Tables 2 and 3).
>
> - **MoNL vs. Per-Modality Training**: MoNL goes beyond per-modality training in UniDiffuser [3], which uses variable noise between modalities i.e., learning $p_{\mathbf{\theta}}([\mathbf{z} _{t ^{(1)}-1} ^{(1)}, \ldots, \mathbf{z} _{t ^{(M)}-1} ^{(M)}] \mid[\mathbf{z} _{t ^{(1)}} ^{(1)}, \ldots, \mathbf{z} _{t ^{(M)}} ^{(M)}])$. MoNL introduces variable noise across different **time segments**, learning connections across **temporal dynamics** as well. This advantage is demonstrated in Table 1.
>
> - **MoNL vs. Masked Training** [4]: Diffusion models often obscure high-frequency details with low noise and low-frequency structures with high noise [5]. MoNL employs variable noise levels to explore diverse **frequency components**, enhancing the model's ability to correlate high and low-frequency elements. This is in contrast to masked self-supervised learning, which limits frequency-specific connections by masking entire elements.
>
> [2] Ruan et al. *"MM-Diffusion: Learning Multi-Modal Diffusion Models for Joint Audio and Video Generation."* CVPR 2023.
>
> [3] Bao et al. *"One Transformer Fits All Distributions in Multi-Modal Diffusion at Scale."* ICML 2023.
>
> [4] Voleti et al. *"MCVD: Masked Conditional Video Diffusion for Prediction, Generation, and Interpolation."* NeurIPS 2022.
>
> [5] Sander Dieleman. *"Noise Schedules Considered Harmful."* [Link](https://sander.ai/2024/06/14/noise-schedules).
>
> $ $
> ### Conclusion
>
> In summary, the effectiveness of **MoNL** for **multimodal diffusion models**, particularly with **audio** and **video** data, stems from its strategic use of **connection** and **heterogeneity**. By applying **variable noise levels**, MoNL enhances **connectivity** between modalities and better adapts to diverse **temporal** and **frequency components**, leading to superior performance compared to existing multimodal learning methods.

---

> ### Author Response · Authors · 2024-08-12
> **Request for Final Review: Response and Theoretical Clarifications**
>
> Dear Reviewer bNvz,
>
>
> We kindly request that you review our responses, as we have only **two days** left for further **discussion**. We have addressed your comments thoughtfully and provided a thorough **theoretical background** of our method. We sincerely appreciate the time and effort you have dedicated to reviewing our paper and your constructive and insightful feedback.
>
> Thank you once again.
>
> Best regards,
> The Authors

---

> ### Author Response · Authors · 2024-08-13
>
> Dear Reviewer bNvz,
>
> We would like to gently remind you that we have **only one day remaining** for further discussion on our manuscript. We have carefully considered your valuable feedback and provided **theoritical backing** for our methodology.
>
> We greatly appreciate your time and insights thus far.
>
> Thank you again for your contributions.
>
> Sincerely,
>
> The Authors

---

> > ### Comment · Reviewer_bNvz · 2024-08-13
> >
> > Dear Authors,
> >
> > I thank you for the detailed explanations and clarifications. After careful consideration and going through the other reviews I’m increasing my rating to weak Accept. The reason of not giving a higher score is due to lack of theoretical novelty.

---

> ### Author Response · Authors · 2024-08-14
>
> Dear Reviewer bNvz,
>
> We sincerely appreciate your time and thoughtful feedback. We are grateful for your recognition of the detailed explanations and clarifications provided, and for upgrading our paper to a "weak accept."
>
> We recognize the need for stronger theoretical foundations. While we acknowledge the current limitations in the theoretical landscape of multimodal learning as highlighted in *"A Theory of Multimodal Learning."*  (Zhou et al. NeurIPS 2023), we believe our research makes a substantial empirical contribution to this field. Our findings serve as a robust foundation for future theoretical explorations and advancements.
>
> We are committed to addressing the theoretical aspects of our work in greater depth in our ongoing research. Thank you once again for your valuable insights.
>
> Sincerely,
>
> The Authors

---

### Author Rebuttal · Authors · 2024-08-07

We sincerely thank the reviewers for their constructive and insightful feedback.

Their recognition of the paper's strengths has been invaluable. We are particularly grateful for their positive comments on:

- **The effectiveness of our approach**: The reviewers highlighted the innovative use of mixed noise levels, which significantly enhances the model's robustness and overall performance.
- **The comprehensive evaluation**: Our fair and thorough experimental analysis, including qualitative and quantitative assessments, has been acknowledged as crucial in demonstrating the method's efficacy.
- **The clarity and readability of the paper**: The reviewers commended the paper's well-structured presentation and ease of understanding.
- **The flexibility and versatility of our model**: The ability to handle diverse audiovisual generation tasks within a unified framework has been recognized as a key contribution.
-  **The strong empirical results**: The reviewers noted the impressive performance of our method across various benchmarks and tasks, including the notable improvements in AV-inpaint and AV-continue.

We have carefully considered all reviewer comments and have made substantial revisions to the paper accordingly. Detailed responses to each point can be found in the rebuttal.

---

### Author Response · Authors · 2024-08-12
**Request for Final Review: Response and Theoretical Clarifications**

Dear Reviewers,

We kindly request that you review our responses, as we have only **two days** left for further **discussion**. We have addressed your comments thoughtfully and provided a thorough **theoretical background** of our method, which will be incorporated in the final version. We sincerely appreciate the time and effort you have dedicated to reviewing our paper and your constructive and insightful feedback.

Thank you once again.

Best regards,
The Authors

---

### Decision · Program_Chairs · 2024-09-25

**Decision:**

Accept (poster)

**Comment:**

This paper proposes to use varying noise levels across modalities and time segments towards task agnostic audio-visual synthesis. The paper received four reviewers; before the rebuttal, the reviewers had raised several concerns on multiple aspects, including technical novelty, gaps in experimental studies, datasets used for experiments, and connections to prior works. Authors provided a strong rebuttal, addressing the concerns, resulting in all the reviewers raising their scores inclining towards acceptance.

AC agrees with the reviewers' assessment of the paper and recommends acceptance. Authors should incorporate all the changes suggested by the reviewers in the camera-ready, including theoretical connections and clarifications from prior methods. Further, as promised, authors should make arrangements for the release of the Monologues dataset.